# Multi-component reactive transport in near-saturated deformable porous media

Bolin Wang<sup>1,2</sup> and Dong-Sheng Jeng <sup>1,2</sup>

<sup>1</sup>College of Civil Engineering, Qingdao University of Technology, Qingdao 266033, China

<sup>2</sup>School of Engineering and Built Environment, Griffith University Gold Coast Campus, Queensland, 4222, Australia

Correspondence: Dong-Sheng Jeng (d.jeng@griffith.edu.au)

Abstract. This study develops a hydro–mechanical–chemical (HMC) framework for simulating reactive solute migration in near-saturated, deformable porous media. The model couples the pore-water mass balance, force equilibrium, and advection–dispersion equations, and further incorporates a flexible geochemical reaction module to address both single-reaction and multi-component, multi-mineral systems. Numerical results indicate that deformation, mechanical loading, saturation, and mineral reactions jointly control the distribution and evolution of the solute. Compression and stronger mechanical loads accelerate solute transport in the early stage but later hinder migration as the pore structure tightens. Moreover, reduced saturation promotes concentration build-up by enhancing advective transport and limiting the ability of the aqueous phase to dilute accumulated solutes. The framework improves the predictive capability for long-term plume behaviour and mineral alteration in reactive porous systems where mechanical, hydraulic, and geochemical processes interact.

## 10 Short Summary

Understanding how reactive solutes move through nearly saturated soil is important for natural environments and engineering practice. However, this movement is influenced by both soil deformation and chemical reactions, which are often treated separately. This paper proposes a flexible framework that links these coupled effects. The results show that changes in loading, moisture, and mineral reactions can shift both the rate and pattern of solute movement, improving long-term predictive capability.

#### 1 Introduction

Solute migration in subsurface environments governs the redistribution of dissolved species within groundwater and other porous formations, influencing water quality, contaminant containment, and the performance of subsurface storage or isolation systems. A clear understanding of these transport processes is crucial for predicting long-term contaminant behaviour and evaluating the reliability of engineered and natural barriers in diverse environmental and industrial contexts.

To better understand the migration of the solute, the first step is to describe the advective and dispersive components of transport. The advection–dispersion equation (ADE), derived from mass conservation under the assumption that hydrodynamic dispersion follows Fick's law (Bear, 1972; Ding et al., 2025; Erfani et al., 2021; Guleria and Chakma, 2023; Xie et al., 2018),

| List of Symbols           |                                                                         |                    |                                                    |  |  |
|---------------------------|-------------------------------------------------------------------------|--------------------|----------------------------------------------------|--|--|
| $S_r$                     | degree of saturation                                                    | n                  | soil porosity                                      |  |  |
| β                         | compressibility of pore water Pa <sup>-1</sup>                          | p                  | excess pore water pressure Pa                      |  |  |
| $\mathbf{v_s}$            | velocity of solid particles m/s                                         | g                  | gravitational acceleration m/s <sup>2</sup>        |  |  |
| K                         | hydraulic conductivity m/s                                              | $K_x$              | <b>K</b> in the <i>x</i> -direction m/s            |  |  |
| $K_y$                     | <b>K</b> in the y-direction m/s                                         | $K_z$              | <b>K</b> in z-direction m/s                        |  |  |
| $\mathbf{u}_{\mathbf{s}}$ | soil displacement m                                                     | $u_x$              | $\mathbf{u}_{s}$ in the x-direction m              |  |  |
| $u_y$                     | $\mathbf{u}_{\mathbf{s}}$ in the y-direction m                          | $u_z$              | $\mathbf{u}_{s}$ in the z-direction m              |  |  |
| $k_{wo}$                  | bulk modulus of pore water MPa                                          | $r_h$              | volumetric fraction of air dissolved in pore water |  |  |
| $p_a$                     | apparent pressure kPa                                                   | $p_0$              | atmospheric pressure kPa                           |  |  |
| $\mu$                     | lamé parameter Pa                                                       | λ                  | lamé parameter Pa                                  |  |  |
| E                         | elastic modulus Pa                                                      | G                  | shear modulus Pa                                   |  |  |
| I                         | unit tensor                                                             | ν                  | Poisson's ratio                                    |  |  |
| $\mathbf{Q}(\mathbf{t})$  | external loading Pa                                                     | $K_d$              | partitioning coefficient $m^3/kg$                  |  |  |
| $ ho_w$                   | density of pore fluid kg/m <sup>3</sup>                                 | $ ho_s$            | density of solid phase kg/m <sup>3</sup>           |  |  |
| $\mathbf{v_f}$            | velocity of fluid particles m/s                                         | $lpha_L$           | longitudinal dispersion coefficient m              |  |  |
| $\alpha_{TH}$             | transverse dispersion coefficient m                                     | $lpha_{TV}$        | transverse dispersion coefficient m                |  |  |
| $D_m$                     | molecular diffusion coefficient m <sup>2</sup> /s                       | $ ar{\mathbf{v}} $ | modulus length of the velocity m/s                 |  |  |
| D                         | hydrodynamic dispersion coefficient m <sup>2</sup> /s                   | Υ.                 | aqueous phase concentration mol/L                  |  |  |
| $Y_s$ .                   | mineral concentration                                                   | J                  | solute flux mol/ $(m^2 \cdot s)$                   |  |  |
| $A_0$                     | specific surface area m <sup>2</sup> /m <sup>3</sup> <sub>mineral</sub> |                    |                                                    |  |  |

has been the most widely applied framework in the field. Due to its simple form, analytical tractability, and computational efficiency, ADE has been recognised as the baseline tool in both aquifer studies and barrier system assessments (Fiori et al., 2017). Although non-Fickian behaviours, such as non-equilibrium diffusion, sorption/desorption, and long-tailed breakthrough curves, have motivated alternative formulations including multirate mass transfer (MRMT) and random walk models (Fiori et al., 2015), these are generally problems not yet universally adopted. In view of ADE's proven reliability and its compatibility with coupled hydro-mechanical modelling, it is adopted here as a robust baseline for solute-transport analysis.

Beyond purely physical transport, geochemical reactions further affect the migration of dissolved species by altering concentration distributions and species mobility. Adsorption/desorption, ion exchange, mineral dissolution—precipitation and redox transformations can retard or accelerate migration through mass exchange or phase change. Capturing these processes has led to the development of reactive-transport modelling (RTM), which provides a unified description of geochemical—hydrological interactions in porous media (Steefel et al., 2015; Guo et al., 2024). Previous studies such as Yeh and Tripathi (1989); Steefel and MacQuarrie (1996) established the general advection—dispersion—reaction framework for multicomponent systems. In recent decades, several RTM platforms, including PHREEQC, TOUGHREACT, and CrunchFlow, have demonstrated how these

65

concepts can be implemented for long-term practical simulations (Pavuluri et al., 2022; Soulaine et al., 2021; Jang et al., 2017; Kempka et al., 2022).

The coupling of reaction processes with transport introduces additional numerical challenges. Two strategies have been commonly classified: *operator-splitting* (sequential) schemes and *fully implicit* (global) methods (Steefel et al., 2005). Sequential schemes decouple transport and reaction within each time increment, offering modularity and reduced computational cost (Samper et al., 2009; Xu and Pruess, 2001). Fully implicit methods solve the entire transport–reaction system simultaneously (Lichtner, 1985; Nghiem et al., 2004), improving stability and mass conservation but often becoming computationally prohibitive for large-scale multi-dimensional problems due to the size and conditioning of the global Jacobian matrix (Yeh and Tripathi, 1989; Ren et al., 2025). Moreover, many fully implicit implementations are tailored to relatively simple or specific reaction networks. For example, Yevugah et al. (2025) considered only the dissolution–precipitation of halite (NaCl) in rock-salt formations, which limits their flexibility for general multicomponent geochemical systems. These considerations explain the continued prevalence of sequential or partially coupled approaches in many engineering applications.

Despite these advances, most hydro-chemical (HC) frameworks still idealise the porous medium as rigid, and thus neglect deformation-induced changes in transport pathways. Several studies have explored mechanical effects in simplified forms. For example, Roded et al. (2018) examined stress-driven dissolution and its impact on the geometry and permeability of the pores at the pore scale. Abdullah et al. (2024) linked carbonate dissolution with porosity, permeability and stress in a single-continuum model. Kadeethum et al. (2021) combined enriched-Galerkin transport with mixed finite element mechanics to simulate calcite precipitation/dissolution during deformation. Guo and Na (2023) further incorporated deformation with multicomponent reactions, but limited the chemistry to the ternary (H<sub>2</sub>O)–(CO<sub>2</sub>)–(CaCO<sub>3</sub>) system. More recently, Mortazavi and Khoei (2024) developed a fully implicit thermo-hydromechanical-chemical (THMC) scheme, but confined the geochemical module to a single dissolution–precipitation reaction. These efforts still simplify geochemistry to restricted reaction terms and cannot represent general multicomponent reactions in deformable porous frameworks.

The gap aforementioned highlights the need for a modelling strategy that consistently integrates the deformation of porous media with comprehensive geochemical reactions. In the present work, a multi-dimensional *hydro-mechanical-chemical* (HMC) reactive-transport framework is implemented to:

- Extends existing near-saturated flow-transport models by embedding a generalised representation of geochemical reactions capable of handling multi-species.
- 2. Captures coupled deformation, solute migration, and geochemical reactions under near-saturated hydraulic conditions, characteristics that are often oversimplified or ignored in previous models.

By bridging geochemical complexity with hydro-mechanical deformation under near-saturated conditions, the proposed framework advances the predictive capability of HMC reactive transport processes in a broad range of subsurface systems, including but not limited to containment barriers, tailings repositories, engineered covers, and natural clay formations. It offers a mechanistic foundation for evaluating long-term hydro-chemo-mechanical evolution in near-saturated clayey soils, thereby

https://doi.org/10.5194/egusphere-2025-5497 Preprint. Discussion started: 27 November 2025

© Author(s) 2025. CC BY 4.0 License.

EGUsphere Preprint repository

70 supporting more reliable risk assessment and design optimisation across environmental remediation, underground storage, and geohazard mitigation.

The remainder of this paper is organised as follows: §2 summarises the modelling assumptions, simplifications and governing equations; §3 presents the numerical implementation; §4 describes the construction of the numerical model together with the material and boundary parameters; §5 discusses the simulation results, emphasising the effects of deformation and multicomponent geochemical reactions, as well as the sensitivity to different loading and saturation conditions; and §6 summarises the key findings and limitations of the present work.

#### 2 Theoretical model

This section outlines the theoretical framework for reactive solute transport in deformable porous media under near-saturated conditions. It first states the key assumptions and simplifications adopted for a continuum-scale description, then introduces the governing equations for the coupled evolution of fluid flow, mechanical response of the porous skeleton, and migration of dissolved elements.

## 2.1 Assumptions and simplifications

The following assumptions define the physical scope of the framework and indicate how they reduce mathematical and computational complexity while highlighting their limitations.

## 35 A. Continuum-scale formulation

The governing hydro–mechanical–chemical equations are formulated at the continuum (Darcy) scale rather than the pore scale. Each control volume is treated as a representative elementary volume (REV) whose characteristic length greatly exceeds the pore size, allowing spatially averaged variables such as pressure, concentration, porosity, and solid deformation, to be defined. This assumption suits field- or engineering-scale problems in deformable porous media where continuum properties (e.g., permeability, dispersion, stiffness) remain well defined and the solid skeleton can be regarded as a homogenised medium.

It becomes invalid for processes controlled by pore-scale heterogeneity such as localised mineral dissolution or precipitation forming preferential channels, pronounced capillary effects in partially saturated zones, or micro-fracture initiation, where explicit pore-scale or hybrid multi-scale approaches are required (Molins et al., 2014, 2017).

# B. Small-strain elastic skeleton in Eulerian frame

The porous skeleton is modelled as linear-elastic and mechanically isotropic, with governing equations expressed in a Eulerian frame under the small-deformation assumption. Inelastic responses such as plastic yielding, visco-elasticity or damage are neglected, so the formulation applies only where materials remain within their elastic range.

The small-strain approximation allows linearised stress-strain relations and avoids the geometric non-linearity and meshupdating required in large-strain analyses. It is unsuitable for substantial volumetric compression, shear localisation or other https://doi.org/10.5194/egusphere-2025-5497 Preprint. Discussion started: 27 November 2025

© Author(s) 2025. CC BY 4.0 License.





on finite-strain effects. The Eulerian description is convenient when material properties are uniform or weakly heterogeneous, since field variables are tracked on a fixed grid. In cases of strong heterogeneity or random variability, significant local deformation can distort material points relative to the mesh, and a Lagrangian or updated Lagrangian scheme is often preferable to capture material interfaces and discontinuities (Fiori et al., 2015; de Barros and Fiori, 2021).

#### C. Neglect of geochemical feedbacks

The present formulation excludes geochemical feedbacks on flow and mechanical behaviour. This simplification is reasonable, as the reactions considered merely redistribute dissolved elements among the aqueous and solid phases, without affecting the pore geometry. Accordingly, chemical processes neither modify porosity nor are influenced by its variation. Under this assumption, both hydraulic and mechanical behaviours are governed exclusively by stress- or strain-induced compaction.

Feedback becomes important when reactions significantly modify the pore space: dissolution enlarges voids and may create preferential channels, whereas precipitation clogs pore throats and reduces permeability. Such reaction-induced changes in porosity and permeability can exceed those caused by mechanical compaction (Abdullah et al., 2024; Seigneur et al., 2019).

#### D. Near-saturated conditions

The porous medium is assumed to remain nearly saturated. The liquid phase forms a continuous network that governs flow, while the gas phase is treated as isolated, immobile bubbles that do not participate in advection. The degree of saturation is therefore prescribed as a constant rather than solved as a dynamic variable, representing conditions where the liquid phase dominates and no significant air invasion occurs. Under this assumption, capillary effects, gas-phase flow, and the transport of dissolved or volatile elements through the gas phase are excluded from the model. The constant saturation also implies that its potential influence on the reaction surface area or transport coefficients is neglected. This simplification is appropriate for wet, deformable porous media in which the liquid phase remains continuous and the gas phase persists only as immobile bubbles; it becomes invalid when continuous gas pathways develop or when gas dissolution, exsolution, or volatilisation exerts a notable influence (Smits et al., 2011; Zeng et al., 2011).

#### E. Mass conservation in elemental form

Mass conservation is fundamental to any reactive solute transport formulation. Here, the transport equation is written in terms of elemental concentrations rather than individual aqueous species. Because reactions merely redistribute elements between dissolved and solid phases without creating or destroying them, elemental balance inherently satisfies mass conservation and requires no explicit reaction source term. This approach is advantageous for multicomponent systems where many species undergo rapid equilibrium or kinetic reactions, yet the total elemental inventory remains constant. For species-based transport problems, suitable reaction source or sink terms must be included to preserve overall elemental mass balance.

## 2.2 Flow and deformation equations

The governing equations for coupled flow-deformation processes in a saturated porous medium originate from two fundamental principles: mass conservation for the pore-water phase and force equilibrium for the deformable solid skeleton. The former governs the flow of the fluid, while the latter describes the mechanical deformation of the solid matrix. These macroscopic forms are well established for deformable porous media under small-strain conditions (Bear, 1972; Jasak and Weller, 2000; Wu and Jeng, 2023).

All dissolved aqueous elements are transported by the same bulk liquid phase and therefore experience a common flow field. Likewise, the deformation of the porous skeleton responds to the total pore-fluid pressure rather than to the concentrations of individual solutes. Consequently, the flow and deformation equations remain unchanged regardless of the number or type of chemical elements present; only the solute transport equation varies with chemical composition.

The pore-water flow equation (Wu and Jeng, 2023) is expressed as

$$S_r n\beta \frac{\partial p}{\partial t} + S_r \nabla \cdot \mathbf{v_s} = \frac{1}{\rho_w g} \nabla \cdot (\mathbf{K} \cdot \nabla p),$$
 (1)

where porosity n is the volume fraction of the void space and controls fluid storage and transmission,  $\rho_w$  is the density of pore water and g is the gravitational acceleration. The coefficient  $\beta$  represents the effective compressibility of the pore fluid mixture and is written as  $\beta = \frac{S_r}{k_{wo}} + \frac{1 - S_r + r_h S_r}{p_a + p_0}$ . Here, the bulk modulus of water is taken as  $k_{wo} = 2000$  MPa, the volumetric fraction of dissolved air is  $r_h = 0.02$ m, and the atmospheric pressure is  $p_0 = 100$  kPa, while  $p_a$  denotes the apparent capillary pressure.

In (1),  $\mathbf{v_s}$  represents the velocity of the solid skeleton, defined as the time derivative of the displacement vector  $\mathbf{u_s} = (u_x, u_y, u_z)$ ; its components are the displacement rates along the Cartesian axes. The hydraulic conductivity tensor  $\mathbf{K}$  is taken as diagonal, with principal components  $K_x$ ,  $K_y$ , and  $K_z$  aligned with the coordinate axes to represent possible anisotropy of the porous medium.

The deformation equation is formulated using the displacement vector  $\mathbf{u}_{s}$  as the primary variable.

$$\nabla \cdot [\mu \nabla \mathbf{u}_{s} + \mu (\nabla \mathbf{u}_{s})^{T} + \lambda \mathbf{I} \operatorname{tr}(\nabla \mathbf{u}_{s})] - \begin{bmatrix} 0 \\ 0 \\ (1 - n)(\rho_{s} - S_{r}\rho_{w})g\nabla \cdot \mathbf{u}_{s} \end{bmatrix} = \nabla p,$$
 (2a)

$$\lambda = \frac{2G\nu}{1 - 2\nu}, \quad \mu = G. \tag{2b}$$

The body force vector in (2a) accounts for the self-weight of the solid skeleton. Because gravity acts vertically, only the z-component is non-zero. In (2b),  $\lambda$  and  $\mu$  are the Lamé parameters defining the elastic response of the porous skeleton. The shear modulus G characterises resistance to shear deformation, and the Poisson's ratio  $\nu$  describes the lateral contraction under uniaxial loading.



# 2.3 Transport equation

The governing equation for solute migration follows the classical advection–dispersion framework widely used in studies of porous media transport (Smith, 2000; Peters and Smith, 2002; Zhang et al., 2012; Wang and Jeng, 2025). A separate transport equation is solved for each chemical element of interest. As highlighted in §2.1, the elemental formulation ensures that chemical reactions merely redistribute each element between the aqueous and solid phases without creating or destroying it. Therefore, no additional reaction source or sink terms are required to enforce mass conservation. Species-level concentrations can be reconstructed by coupling these elemental balances with reaction relations, but the governing transport operator itself remains the same for all elements.

The solute transport equation (Wang and Jeng, 2025) is written as

$$[S_r n + (1 - n)\rho_s K_d] \frac{\partial Y_s(t)}{\partial t} = S_r n \mathbf{D} : \nabla^2 Y_s(t) + S_r n \nabla Y_s(t) \cdot \nabla \cdot \mathbf{D} - S_r n \mathbf{v_f} \cdot \nabla Y_s(t) - [(1 - n)\rho_s K_d] \mathbf{v_s} \cdot \nabla Y_s(t) + (S_r n \beta \frac{\partial p}{\partial t} + S_r n \beta \mathbf{v_f} \cdot \nabla p) Y_s(t),$$
(3)

$$\mathbf{v_f} = -\frac{1}{\rho_w g S_r n} \mathbf{K} \cdot \nabla p + \mathbf{v_s},\tag{4}$$

In (3), Y.() denotes the dissolved concentrations of aqueous elements, for example Y.C for the total dissolved carbon and Y.Ca for calcium in the liquid phase. Mineral-phase concentrations, such as Ys.Calcitetetetetetetetetetetetetetetetetetetetetetetetetetetetetetetetetetetetetetetetetetetetetetetetetetetetetetetetetetetetetetetetetetetetetetetetetetetetetetetetetetetetetetetetetetetetetetetetetetetetetetetetetetetetetetetetetetetetetetetetetetetetetetetetetetetetetetetetetetetetetetetetetetetetetetetetetetetetetetetetetetetetetetetetetetetetetetetetetetetetetetetetetetetetetetetetetetetetetetetetetetetetetetetetetetetetetetetetetetetetetetetetetetetetetetetetetetetetetetetetetetetetetetetetetetetetetetetetetetetetetetetetetetetetetetetetetetetetetetetetetetetetetetetetetetetetetetetetetetetetetete

The hydrodynamic-dispersion tensor **D** is defined using the local-scale Scheidegger model, which treats dispersion as the combined effect of molecular diffusion and velocity-dependent mechanical dispersion under a spatially smooth Darcy-scale velocity field.

$$D_{xx} = \alpha_L \frac{v_x^2}{|\bar{\mathbf{v}}|} + \alpha_{TH} \frac{v_y^2}{|\bar{\mathbf{v}}|} + \alpha_{TV} \frac{v_z^2}{|\bar{\mathbf{v}}|} + D_m, \quad D_{yy} = \alpha_{TH} \frac{v_x^2}{|\bar{\mathbf{v}}|} + \alpha_L \frac{v_y^2}{|\bar{\mathbf{v}}|} + \alpha_{TV} \frac{v_z^2}{|\bar{\mathbf{v}}|} + D_m, \tag{5}$$

$$D_{zz} = \alpha_{TV} \frac{v_x^2}{|\overline{\mathbf{v}}|} + \alpha_{TV} \frac{v_y^2}{|\overline{\mathbf{v}}|} + \alpha_L \frac{v_z^2}{|\overline{\mathbf{v}}|} + D_m, \quad D_{yx} = D_{xy} = (\alpha_L - \alpha_{TH}) \frac{v_x v_y}{|\overline{\mathbf{v}}|}, \tag{6}$$

$$D_{xz} = D_{zx} = (\alpha_L - \alpha_{TV}) \frac{v_x v_z}{|\bar{\mathbf{v}}|}, \quad D_{yz} = D_{zy} = (\alpha_L - \alpha_{TV}) \frac{v_y v_z}{|\bar{\mathbf{v}}|}. \tag{7}$$

Here, the molecular diffusion coefficient  $D_m$  accounts for Brownian-motion-driven spreading in the absence of bulk flow. The longitudinal and transverse mechanical-dispersion coefficients  $\alpha_L$ ,  $\alpha_{TH}$  and  $\alpha_{TV}$  describe velocity-dependent spreading caused by microscopic variations in pore-scale flow paths. The modulus length of the velocity vector  $|\bar{\mathbf{v}}|$  is calculated from



$$|\bar{\mathbf{v}}| = \sqrt{(v_{fx} - v_{sx})^2 + (v_{fy} - v_{sy})^2 + (v_{fz} - v_{sz})^2} = \frac{1}{\rho_w g S_r n} \sqrt{\left(-K_x \frac{\partial p}{\partial x}\right)^2 + \left(-K_y \frac{\partial p}{\partial y}\right)^2 + \left(-K_z \frac{\partial p}{\partial z}\right)^2},$$
 (8)

where  $v_{fx}$ ,  $v_{fy}$ ,  $v_{fz}$  and  $v_{sx}$ ,  $v_{sy}$ ,  $v_{sz}$  are the fluid and solid velocity components in the x-, y-, and z-directions, respectively.

Equation (3) includes a simple linear adsorption term represented by the distribution coefficient  $K_d$  to model instantaneous equilibrium partitioning between the aqueous phase and the solid matrix. If a more detailed description of adsorption—desorption kinetics or competitive sorption among multiple elements is needed, this linear term can be removed and replaced by reaction-controlled source—sink terms calculated by the geochemical reaction module, which updates both aqueous and solid-phase concentrations.

$$S_r n \frac{\partial Y_r(t)}{\partial t} = S_r n \mathbf{D} : \nabla^2 Y_r(t) + S_r n \nabla Y_r(t) \cdot \nabla \cdot \mathbf{D} - S_r n \mathbf{v_f} \cdot \nabla Y_r(t) + (S_r n \beta \frac{\partial p}{\partial t} + S_r n \beta \mathbf{v_f} \cdot \nabla p) Y_r(t). \tag{9}$$

#### 3 Numerical scheme

#### 3.1 Development of custom solvers

# 195 A. Numerical platform

The numerical framework is developed on the open-source finite-volume platform (OpenFOAM v8), which provides robust solvers for continuum-scale flow, deformation and solute transport processes, and offers high parallel scalability for large multi-dimensional problems. Geochemical reactions are handled by PhreeqcRM, the reaction module interface of Parkhurst and Wissmeier (2015), which preserves the full thermodynamic and kinetic reaction capabilities of PHREEQC while streamlining data exchange between the reactive-transport code and the geochemical engine. The coupling strategy follows the concept of the porousMedia4Foam project (Pavuluri et al., 2022), but the present architecture is further optimised and extended to account for deformable porous media under near-saturated conditions.

The coupled hydro-mechanical-chemical (HMC) system is organised into three modules:

- Hydro-mechanical (HM) module OpenFOAM: solves the groundwater flow and solid-deformation equations, yielding the transient pore pressure field and the displacement field of the porous skeleton.
- Transport module OpenFOAM: solves the advection-dispersion equations for each chemical element to update its
  concentration field across the computational domain.
- Chemical-reaction module PhreeqcRM: computes equilibrium and kinetic reactions based on element concentrations supplied by OpenFOAM and returns the updated elemental composition to maintain the general mass balance.



The above three modules are integrated using an operator-splitting scheme (Lie or Strang), which provides a practical balance between numerical accuracy and computational efficiency. This approach avoids the high cost of a fully implicit global solution while enabling stable and scalable HMC simulations for complex multi-dimensional domains.

# B. Treatment of molecular diffusion in the element-based transport framework

The definition of the molecular diffusion coefficient  $D_m$  for element-based transport is inherently ambiguous. Each chemical element may exist in multiple aqueous species (for example,  $Ca^{2+}$  and  $CaHCO_3^+$ ), which often have markedly different diffusion coefficients and charge states. Furthermore,  $D_m$  depends on temperature, ionic strength, and local chemical composition, all of which can vary spatially and temporally in reactive systems. Consequently, defining an element-level diffusion coefficient requires a simplifying assumption.

In this study, a generalised  $D_m$  is calculated as a weighted average of the main aqueous species to ensure both simplicity and flexibility of the framework. Let  $\omega_i$  denote the user-defined fraction of the element present in aqueous elements i ( $0 \le \omega_i \le 1$ ,  $\sum_{i=1}^{N_s} \omega_i = 1$ ). The effective coefficient is then defined as

$$D_m = \sum_{i=1}^{N_s} \omega_i D_m^{(i)},\tag{10}$$

where  $D_m^{(i)}$  is the molecular diffusion coefficient of the elements i. The solver reads  $\omega_i$  and  $D_m^{(i)}$  from an external .txt file, provided as constants or as spatial- or time-dependent fields. If no fractions are supplied, a single constant  $D_m$  specified by the user is applied throughout the domain. For the case studies presented later, one involving a single mineral reaction system and the other a multi-mineral network, the full information of the aqueous elements and their assigned diffusion coefficients are provided in Appendix A.

Although the weighted average definition of  $D_m$  provides a practical and flexible means to represent element-based diffusion, it relies on one key simplifying assumption. The formulation neglects electromigration and other multicomponent coupling effects that are explicitly resolved in the Nernst–Planck equation and the Maxwell–Stefan diffusion theory (Bard et al., 2022; Krishna and Wesselingh, 1997). Consequently, the present framework can serve as a foundational platform for future developments, where multicomponent diffusion can be incorporated to account for electromigration effects. Such extensibility allows the model to accommodate more complex chemical environments without changing its numerical structure.

## C. Reaction activation window

The framework enables users to activate or deactivate the geochemical module by defining time windows in an external configuration file. During inactive periods, the solver advances only the hydro-mechanical and transport modules, thereby reducing computational overhead and improving overall efficiency. Within each activation window, the treatment of the reactions depends on their type. When only equilibrium reactions are considered, a single chemical calculation is performed at the begin-

ning of the activation period to update the system to equilibrium. For kinetic reactions, however, PhreeqcRM subdivides each transport time step into multiple reaction sub-steps to resolve the reaction dynamics in greater detail.

In this framework (11), the solute transport process defines the main clock that governs the overall time progress of the simulation, while chemical reactions evolve in an internal sub-clock within each transport step. Each transport interval of duration  $\Delta t_{\text{transport}}$  is therefore subdivided into smaller chemical sub-steps  $\Delta t_{\text{reaction}}$ , such that  $\Delta t_{\text{transport}} = N_r \Delta t_{\text{reaction}}$ . See B for a detailed description of the chemical activation window.

Although the dual-clock scheme allows chemical reactions to evolve on finer internal sub-steps, it also introduces synchronisation issues. Extremely small transport steps are required to capture fast kinetics, which may offset the intended efficiency gain, whereas overly large sub-steps can reduce temporal resolution and cause numerical errors. Nevertheless, this framework is well suited for problems where the transport and reaction processes operate on distinct time scales, and the chemical feedback is relatively weak, providing an effective balance between accuracy and computational efficiency.

# 3.2 Integration strategy

The overall integration workflow is illustrated in Figure 1. At the start of the simulation, PhreeqcRM performs an initialisation stage that reads the configuration parameters, loads the prescribed geochemical conditions, and computes the initial aqueous-element concentrations and mineral-phase abundances. In coordination with OpenFOAM, these initial values are mapped onto the computational mesh so that each control volume is assigned its starting aqueous and mineral concentrations. These fields serve as the baseline for subsequent transport and reaction calculations.

For each global time step thereafter, OpenFOAM first solves the hydro-mechanical (HM) subsystem, pore pressure, mechanical deformation and elemental advection—dispersion transport. Although the flow and deformation equations are strongly coupled through the pore-pressure—displacement interaction, solving them in a fully monolithic manner is numerically demanding and often computationally expensive. To improve robustness and efficiency, the present framework adopts a segregated iterative strategy, which has been widely used in CFD. The pressure equation is solved first using the previous iteration's displacement field. The resulting pressure distribution is then used to update the solid skeleton deformation, and only explicit terms are refreshed at each iteration. This process repeats until the change between successive iterations falls below a pre-

scribed tolerance, at which point convergence is declared. The convergent fields of pore pressure p and displacement  $\mathbf{u_s}$  are then passed in a one-way manner to the transport module to define the pore-fluid velocity  $\mathbf{v_f}$  and solid velocity  $\mathbf{v_s}$  required by the advection–dispersion equation for solute migration.

After the HM and transport updates, the solver checks whether the current simulation time falls within the user-defined reaction-activation window. If the chemical module is inactive, the solver skips the reaction step and proceeds directly to the next global time step. If the chemical module is active, OpenFOAM first assembles the elemental concentration fields from all computational cells into a contiguous vector conforming to the data structure required by PhreeqcRM. This vector is passed to PhreeqcRM, which performs aqueous speciation and kinetic-reaction calculations to update both aqueous concentrations and mineral-phase abundances. Upon completion, PhreeqcRM overwrites the same vector with the updated results, which are then mapped back into the OpenFOAM fields of each cell. The simulation proceeds to the next global HM–transport step using these updated concentrations.

# 275 3.3 Verification with previous studies

Two benchmark problems were used to verify different components of the present model and to examine the numerical stability of the solvers. Laboratory data is not available in the literature for direct comparison. However, benchmarking against published numerical results is a well-established practice in the field of reactive solute transport, and provides a practical means to confirm model implementation before applying it to new scenarios.

## 280 Verification #1: Hydro-mechanical-transport (without reactions)

The first benchmark was designed to validate the hydro–mechanical–transport (HM–T) module implemented in OpenFOAM, with chemical reactions excluded. All material properties, boundary conditions, and loading settings were kept identical to those reported by Wu and Jeng (2023), who solved the same problem using the commercial package COMSOL.

The dissipation of the simulated pore pressure and vertical displacement show excellent agreement with Wu and Jeng (2023) in the domain and in all reported directions, confirming the precision of the coupled flow–deformation implementation (Figure 2). A minor early-time deviation appears in the breakthrough concentration at the model top. A plausible cause for this discrepancy is the difference in dimensional treatment between the two codes: COMSOL solves a strictly planar (2-D) problem, whereas OpenFOAM emulates 2-D by extruding a thin 3-D slab.

This quasi-2-D setup introduces several non-ideal effects: (i) the finite out-of-plane thickness implies a small but non-zero storage capacity, which can alter the volumetric flux per unit width; (ii) the high-aspect-ratio cells in the thin direction tends to increase the numerical (and possibly physical) transverse dispersion, which slightly smears the sharp concentration front; (iii) differences in boundary treatment for the thin direction (e.g., empty vs. symmetryPlane) and related gradient corrections can produce small phase shifts in the concentration profile.

Figure 1. Numerical simulation procedure for present model.

Figure 2. Comparison between the present model and previous model (Wu and Jeng, 2023). (a) Pressure, (b) vertical displacement and (c) solute concentration  $c_f$  vs. time. Note: red = blue line indicates the top; red line indicates the middle; black line indicates the bottom.

**Figure 3.** Comparison of the present model (solid line) with Benchmark 2 (Pavuluri et al. (2022), diamond markers). The blue line represents the result at 20 minutes, the red line at 40 minutes, and the black line at 60 minutes.

305

## **Verification #2: Reactive solute transport with kinetic dissolution**

The second benchmark was designed to evaluate the OpenFOAM–PhreeqcRM coupling for simulating kinetically controlled mineral reactions under advective–dispersive transport. In this test, the HM module was disabled and a constant velocity of  $1 \times 10^{-4}$  m<sup>3</sup>/s (volumetric injection rate) was imposed. The solid matrix comprised 57 % inert mineral and 3 % calcite, while an acid solution of pH = 2 was continuously injected, following the configuration reported by Pavuluri et al. (2022) (also implemented in OpenFOAM). A one-dimensional transport domain was used to represent calcite dissolution and precipitation during migration.

The simulated concentration profiles of dissolved Ca were compared with those of the reference study, as illustrated in Figure 3. These two solutions agree well at intermediate and late times, but noticeable deviations appear in the early stage of the breakthrough. These differences arise primarily from variations in the underlying transport formulations and boundary representations. The present solver accounts for near-saturation effects, which modify the advective flux through compressibility terms, whereas the reference benchmark adopted a classical ADE-based formulation with constant porosity and a prescribed uniform flow field. As the dissolution front advanced and concentration gradients became less steep, the two solutions gradually converged, although a small residual discrepancy persisted because of the fundamentally different governing equations and kinetic parameterisations.

# 4 Application to a landfill liner

The developed framework is designed to simulate coupled flow, deformation and multicomponent reactive transport in porous media subjected simultaneously to mechanical loading and chemically aggressive fluids. This modelling tool is broadly applicable to scenarios where barrier materials interact with reactive solutes, for example, chloride ingress that accelerates steel corrosion in reinforced concrete structures (Guo et al., 2024), or advective–dispersive transport through clay liners accompanied by cation exchange and carbonate dissolution or precipitation (Guo and Na, 2023).

In this study, the framework is applied to a landfill-liner scenario. Chemically aggressive leachates, such as acidic solutions enriched in  $CO_3^-$ ,  $SO_4^{2-}$ , can infiltrate the liner and react with both its mineral constituents and the underlying geomaterials, thus improving contaminant migration and potentially compromising the long-term performance of the barrier.

## 4.1 Model description

In OpenFOAM, a quasi-2D domain (Figure 4) was built as a thin 3D slab of length  $L_x = 20$  m and height  $L_z = 1$  m, with a negligible out-of-plane thickness to emulate two-dimensional behaviour. Along the top boundary, three equal-length contaminant inlets (depicted by blue arrows) are separated by source-free segments, whereas the remaining top surface is non-reactive with respect to solute input and is subjected to a uniform mechanical load. The bottom boundary comprises five localised drainage outlets (red arrows) with a prescribed hydraulic head, with the rest rendered impermeable; both lateral boundaries are also impermeable and mechanically fixed to prevent rigid body motion.

Figure 4. Cross section of the present model.

The coordinate system is defined such that *x* is positive to the right and *z* is positive upward. The geochemical initial condition is spatially uniform: the entire computational domain is assigned Solution 1 from the PHREEQC input file, thus providing the same initial aqueous composition throughout the liner and the surrounding geomaterial. Each OpenFOAM cell is directly mapped to a corresponding PHREEQC cell, ensuring a one-to-one correspondence between the flow and geochemical grids. If spatially variable initial compositions are required (i.e., different Solution definitions), the OpenFOAM mesh must be preconfigured to assign the appropriate PHREEQC solution number to each computational cell.

## 4.2 Boundary conditions and input parameters

A mechanical load is applied at the top boundary and expressed through the local force-balance condition.

$$-\mathbf{n} \cdot \boldsymbol{\sigma}' = \mathbf{Q}(t) - \mathbf{n} \cdot p \mathbf{I},\tag{12}$$

where **n** is the outward unit normal,  $\sigma'$  denotes the effective stress tensor, and **I** is the identity tensor. The effective stress is defined following Jasak and Weller (2000), and the corresponding displacement gradient on the loaded surface is given by

$$\mathbf{n} \cdot \nabla \mathbf{u}_{s} = \frac{-\mathbf{Q}(t) - [\mathbf{n} \cdot \mu(\nabla \mathbf{u}_{s})^{T} + \lambda \mathbf{I} \operatorname{tr}(\nabla \mathbf{u}_{s}) - (\mu + \lambda) \nabla \mathbf{u}_{s}] + \mathbf{n} p}{2\mu + \lambda}.$$
(13)

The bottom boundary is constrained to emulate a stiff foundation: both vertical and lateral components of displacement are fixed at zero, suppressing downward settlement and horizontal sliding along the base. The lateral boundaries are laterally fixed to prevent rigid-body motion of the entire domain, but their vertical displacement remains free unless otherwise specified.

For pore pressure, the top surface is assumed to be in contact with an overlying ponded leachate at atmospheric pressure and is therefore assigned a zero normal gradient condition,

$$\frac{\partial p(x, y, 0, t)}{\partial z} = 0,\tag{14}$$

which is consistent with a flat free surface in static equilibrium and negligible vertical seepage. Under near-saturated conditions, the liquid phase remains continuous up to the surface while the gas phase is disconnected, so no capillary head or hydrostatic




jump develops across the interface. If a sustained head difference develops between the ponded leachate and the liner interior, a non-zero gradient or a Dirichlet-type pressure specification would be required to represent advective inflow.

The lateral boundaries are treated as impermeable for flow, enforcing no-drainage (zero-normal-flux) conditions. At the bottom boundary, drainage is permitted only at the designated outlet segments where hydraulic head is prescribed, while the remaining sections of the bottom share the same no-drainage condition as the lateral boundaries.

The boundary of solute concentration at the top is specified as

$$\frac{\partial Y_{\cdot}(x,y,0,t)}{\partial z} = -\frac{J^{(i)}}{S_{r}nD_{m}^{(i)}},\tag{15}$$

where  $J^{(i)}$  the imposed vertical solute-mass flux of elements i, and  $D_m^{(i)}$  the molecular diffusion coefficient. This Robin-type, diffusion-controlled boundary is appropriate when the overlying leachate pond has negligible vertical seepage yet maintains a concentration gradient that drives diffusive mass transfer into the liner. If appreciable advective inflow occurs, a prescribed advective flux or a Dirichlet-type concentration boundary would be more suitable. The lateral boundaries and the impermeable portions of the bottom boundary are assigned zero normal gradient (Neumann) conditions for concentration. The boundary conditions are summarised in Table 1.

**Table 1.** Boundary conditions (BCs) setup for the present models.

| Boundary Index               | BC of p                             | BC of u <sub>s</sub>                                                            | <b>BC</b> of <i>Y</i> .()                                             |
|------------------------------|-------------------------------------|---------------------------------------------------------------------------------|-----------------------------------------------------------------------|
| Top (With pollution sources) | $\frac{\partial p}{\partial z} = 0$ | $\mathbf{n} \cdot \boldsymbol{\sigma}' = -\mathbf{Q}(\mathbf{t}) + p\mathbf{n}$ | $\frac{\partial Y(t)}{\partial z} = -\frac{J^{(i)}}{S_T n D_m^{(i)}}$ |
| Top (No pollution sources)   | $\frac{\partial p}{\partial z} = 0$ | $\mathbf{n} \cdot \boldsymbol{\sigma}' = -\mathbf{Q}(\mathbf{t}) + p\mathbf{n}$ | $\frac{\partial Y_{\cdot}()}{\partial z} = 0$                         |
| Bottom (Drainage area)       | p = 0                               | $\mathbf{u}_{\mathbf{s}} = 0$                                                   | $\frac{\partial Y(t)}{\partial z} = 0$                                |
| Bottom (Undrained area)      | $\frac{\partial p}{\partial z} = 0$ | $\mathbf{u_s} = 0$                                                              | $\frac{\partial Y(t)}{\partial z} = 0$                                |
| Sides                        | $\frac{\partial p}{\partial x} = 0$ | $\mathbf{u_s} = 0$                                                              | $\frac{\partial Y(t)}{\partial x} = 0$                                |

The simulated liner is modelled as a low permeability clayey soil under a small-strain linear elastic framework (Table 2). Hydraulic and mechanical properties—including porosity n, degree of saturation  $S_r$ , hydraulic conductivity  $\mathbf{K}$ , shear modulus G, Poisson's ratio v, and compressibility coefficients—were taken from representative ranges for compacted clay liners (Peters and Smith, 2002; Zhang et al., 2012). These values are consistent with the small-strain assumption, allowing the use of linear elasticity and the Eulerian formulation without large-deformation corrections.

For reactive-transport simulations, the linear distribution coefficient  $K_d$ , a simplified equilibrium sorption term, was set to zero for all elements so that changes in breakthrough and spatial distribution arise solely from geochemical reactions such as mineral dissolution and precipitation. The reaction module (PhreeqcRM) can also handle more complex sorption processes, including nonlinear (Freundlich or Langmuir) and competitive or kinetic adsorption—desorption, if specified in the chemical database.

**Table 2.** Physical parameters of the present models.

| Parameter       | Value                                         | Description                               |  |  |
|-----------------|-----------------------------------------------|-------------------------------------------|--|--|
| *S <sub>r</sub> | 0.88                                          | Degree of saturation                      |  |  |
| n               | 0.33                                          | Soil initial porosity                     |  |  |
| *Q(t)           | Referring to Figure 5                         | External load                             |  |  |
| G               | $2.75 \times 10^{6} \mathrm{Pa}$              | Shear modulus                             |  |  |
| ν               | 0.33                                          | Poisson's ratio                           |  |  |
| $K_x$           | $1 \times 10^{-10} \mathrm{m/s}$              | Hydraulic conductivity in the x direction |  |  |
| $K_z$           | $1 \times 10^{-11} \mathrm{m/s}$              | Hydraulic conductivity in the z direction |  |  |
| $ ho_w$         | $1 \times 10^3 \mathrm{kg/m^3}$               | Density of the pore fluid                 |  |  |
| $ ho_s$         | $2.6 \times 10^3 \mathrm{kg/m^3}$             | Density of the soil                       |  |  |
| g               | $9.8 \text{ m/s}^2$                           | Gravitational acceleration of water       |  |  |
| $K_d$           | $0\mathrm{m}^3/\mathrm{kg}$                   | Partitioning coefficient                  |  |  |
| $lpha_L$        | 0.5 m                                         | Longitudinal dispersion coefficient       |  |  |
| $\alpha_T$      | 0.1 m                                         | Transverse dispersion coefficient         |  |  |
| $A_0$           | $0.1 \text{ m}^2/\text{m}_{\text{mineral}}^3$ | Specific surface area of mineral          |  |  |

Note: Parameters with \* were used in the parametric study.

## 4.3 Description of case studies

Two types of reactive-transport scenarios were designed to demonstrate and evaluate the capabilities of the proposed HMC framework. The first case considers a single reactive mineral that undergoes multicomponent aqueous reactions, enabling a clear examination of the fundamental coupling between hydro-mechanical processes and geochemical reaction fronts. This case is also used to explore the sensitivity of the system response to external mechanical loading and to variations in the initial degree of saturation, highlighting the interaction between pore pressure dissipation, deformation, and reactive solute migration.

The second case extends the system to a multi-mineral, multi-component reaction network to showcase the stability of the framework to capture simultaneous dissolution—precipitation and competitive aqueous reactions among several solid phases. This progression from a simplified to a more complex geochemical setting highlights the versatility of the solver and its ability to handle general reactive transport problems.

## A. Single mineral reaction


The dissolution and precipitation of calcite in the CO<sub>2</sub>-water system follow the classical carbonate reaction,

$$CaCO_3(s) + CO_2(aq) + H_2O(1) \rightleftharpoons Ca^{2+}(aq) + 2HCO_3^-(aq),$$
 (16)

which describes the equilibrium between solid calcite, dissolved CO<sub>2</sub>, and bicarbonate species.



Rather than assuming instantaneous equilibrium, calcite dissolution and precipitation were represented using the surface-controlled kinetic law (Plummer et al., 1978):

$$r_{\nu}(t) = A_{\text{eff}}(M) \left[ k_1(T) a_{\text{H}^+} + k_2(T) a_{\text{H}_2\text{CO}_3^*} + k_3(T) a_{\text{H}_2\text{O}} \right] \left( 1 - 10^{\eta S I_{\text{calcite}}} \right), \tag{17}$$

where  $r_v(t)$  is the volumetric rate of reaction (mol m<sub>bulk</sub><sup>-3</sup> s<sup>-1</sup>);  $A_{\text{eff}}(M) = A_0 (M/M_0)^m$  is the effective reactive surface area that scales with the remaining mineral mass M;  $k_1$ ,  $k_2$ , and  $k_3$  are temperature-dependent rate constants representing the acid (H<sup>+</sup>), carbonic-acid (H<sub>2</sub>CO<sub>3</sub>\*), and neutral (H<sub>2</sub>O) mechanisms, respectively;  $a_i$  denotes the activity of aqueous species i;  $SI_{\text{calcite}}$  is the saturation index controlling the departure from equilibrium (SI < 0 favours dissolution, SI > 0 favours precipitation);  $\eta$  is an empirical coefficient (typically  $\eta \approx 1$ ), and m is the surface-area scaling exponent (commonly m = 2/3).

Table 3 lists the primary chemical components of the background pore water and the injected acidic solution. The porous medium initially contained 57 % inert mineral and 3 % calcite. The background water was near-neutral,  $CO_2$ -buffered, and close to calcite saturation, while the injected solution was strongly acidic (pH = 2). Acidic inflow reduced the local calcite saturation index, promoting calcite dissolution and releasing  $Ca^{2+}$  and  $HCO_3^-$  into the aqueous phase. The solute fluxes imposed for inorganic carbon differed between the three injection points, designated  $J_1$ ,  $J_2$  and  $J_3$  as shown in Figure 4. These sources are located in the left, middle, and right sections of the upper boundary, respectively, with the injection strength progressively increasing from  $J_1$  to  $J_3$ .

**Table 3.** Primary components in the background (initial) and injected solutions for the single-mineral reaction.

| Component           | Initial condition                               | Injected solution                                               |  |
|---------------------|-------------------------------------------------|-----------------------------------------------------------------|--|
| pН                  | 8.2                                             | 2.0                                                             |  |
| Ca                  | $1 \times 10^{-3} \text{mol/L}$                 | negligible                                                      |  |
| С                   | $1 \times 10^{-3} \text{mol/L}$                 | $J_1 = 2.5 \times 10^{-10} \text{mol/(m}^2 \cdot \text{s)},$    |  |
|                     |                                                 | $J_2 = 5 \times 10^{-10} \text{mol/(m}^2 \cdot \text{s)},$      |  |
|                     |                                                 | $J_3 = 1 \times 10^{-9} \text{mol}/(\text{m}^2 \cdot \text{s})$ |  |
| CO <sub>2</sub> (g) | $\log_{10} p_{\text{CO}_2} = -3.45 \text{ atm}$ | negligible                                                      |  |

To examine the effects of hydro-mechanical coupling, external loading, and saturation on reactive solute migration, three groups of simulation scenarios were designed:

# 1. Group 1: Mechanistic influence (Case A-C)

- Case A: Baseline full HMC coupling (hydro-mechanical coupling, solute transport, and chemical reactions); vertical load  $4.0 \times 10^5$  Pa applied for two years at  $S_r = 0.88$ .
- Case B: Same as Case A, but with geochemical reactions deactivated (no-chem).
- Case C: Same as Case A, but with the mechanical module removed (no-mech; reactive transport only).


# 2. Group 2: Load sensitivity (Case D-E) (Figure 5)

- Case D: Higher vertical load  $6.0 \times 10^5$  Pa, other settings as in Case A (HMC).
- Case E: Lower vertical load  $2.0 \times 10^5$  Pa, other settings as in Case A (HMC).

# 3. Group 3: Saturation sensitivity and module exclusion (Case F-K)

- Case F: HMC at  $S_r = 1.0$  (other settings as in Case A).
- Case G: HMC at  $S_r = 0.80$  (other settings as in Case A).
- Case H: No-mech (RT+Chem) at  $S_r = 1.0$  (same as Case F but without mechanics).
- Case I: No-mech (RT+Chem) at  $S_r = 0.80$  (same as Case G but without mechanics).
- Case J: No-chem (HM+T) at  $S_r = 1.0$  (same as Case F but without geochemical reactions).
- Case K: No-chem (HM+T) at  $S_r = 0.80$  (same as Case G but without geochemical reactions).

Figure 5. Load pattern under different models: the black line represents Case A, the red line represents Case D, and the blue line represents Case E.

#### **B.** Multi-mineral reaction

The simulations also consider two additional reactive minerals in **Model L**: gibbsite (Al(OH)<sub>3</sub>) and siderite (FeCO<sub>3</sub>), whose dissolution–precipitation reactions in aqueous systems are

$$Al(OH)_3(s) + 3H^+(aq) \rightleftharpoons Al^{3+}(aq) + 3H_2O(l),$$
 (18)

$$FeCO3(s) + H+(aq) \rightleftharpoons Fe2+(aq) + HCO3-(aq),$$
(19)

Both minerals were treated as non-equilibrium phases whose dissolution or precipitation follows a surface-area–controlled kinetic law:

$$r_{A,j}(t) = k_j(T)(1 - 10^{\eta_j S I_j}), \qquad j \in \{\text{Gibbsite, Siderite}\},\tag{20}$$

where  $r_{A,j}$  is the areal rate (mol m<sup>-2</sup> s<sup>-1</sup>),  $k_j(T)$  is a temperature-dependent rate constant (e.g.  $k_{\text{gibbsite}} = 5 \times 10^{-10}$  and  $k_{\text{siderite}} = 5 \times 10^{-9}$  mol m<sup>-2</sup> s<sup>-1</sup> in the implementation of PHREEQC at 25°C),  $SI_j$  is the saturation index of the mineral j and  $\eta_j$  is an empirical coefficient (typically  $\eta_j \approx 1$ ).

Table 4 summarises the multi-mineral scenario, which represents an acidic and oxidising plume injected into a near-neutral, CO<sub>2</sub>-buffered carbonate aquifer. The aquifer initially contained calcite (22 %), gibbsite (5 %), siderite (5 %), and an inert mineral (33 %). The injected plume, characterised by low pH and continuous inflow of C, Fe, and Al, interacted with near-neutral background groundwater and altered its pH and redox state. These chemical perturbations could dissolve or precipitate the resident carbonate and hydroxide minerals, depending on the degree of mixing between the injected plume and the background pore water.

Table 4. Primary components in the background (initial) and injected solutions for the multi-mineral reaction.

| Component             | Initial condition                                 | Injected solution                                                 |  |
|-----------------------|---------------------------------------------------|-------------------------------------------------------------------|--|
| рН                    | 8.0                                               | 3.0                                                               |  |
| Ca                    | 0.01 mol/L                                        | negligible                                                        |  |
| С                     | $1 \times 10^{-3}$ mol/L                          | $J_1 = 2.5 \times 10^{-10} \text{mol/(m}^2 \cdot \text{s)},$      |  |
|                       |                                                   | $J_2 = 5 \times 10^{-10} \text{mol/(m}^2 \cdot \text{s)},$        |  |
|                       |                                                   | $J_3 = 1 \times 10^{-9} \text{mol/}(\text{m}^2 \cdot \text{s})$   |  |
| Fe                    | 0.01 mol/L                                        | $J_1 = 4 \times 10^{-9} \text{mol/(m}^2 \cdot \text{s)},$         |  |
|                       |                                                   | $J_2 = 8 \times 10^{-9} \text{mol/(m}^2 \cdot \text{s)},$         |  |
|                       |                                                   | $J_3 = 1.6 \times 10^{-8} \text{mol}/(\text{m}^2 \cdot \text{s})$ |  |
| Al                    | 0.01 mol/L                                        | $J_1 = 3 \times 10^{-10} \text{mol/(m}^2 \cdot \text{s)},$        |  |
|                       |                                                   | $J_2 = 5.5 \times 10^{-10} \text{mol/(m}^2 \cdot \text{s)},$      |  |
|                       |                                                   | $J_3 = 1.5 \times 10^{-9} \text{mol/(m}^2 \cdot \text{s})$        |  |
| pe/O <sub>2</sub> (g) | pe = 2                                            | pe = 14, $\log_{10} p_{O_2} = -1$ (atm)                           |  |
| CO <sub>2</sub> (g)   | $\log_{10} p_{\text{CO}_2} = -3.45 \text{ (atm)}$ | negligible                                                        |  |

## 5 Numerical results and discussion

This section presents numerical simulations for landfill liner scenarios and highlights the key physical insights from coupled hydromechanical-chemical (HMC) analyses. Subsection 5.1 examines how deformation combined with chemical reac-





tions affects pore pressure dissipation, solute migration, and mineral alteration, and compares the system responses under different external loads. Subsection 5.2 evaluates how changes in the initial degree of saturation modify the coupled hydromechanical—chemical behaviour. Subsection 5.3 investigates the role of multiple reactive minerals in shaping the evolution of concentration and the advancement of reaction fronts. Subsection 5.4 discusses the trade-off between model accuracy and computational cost, providing guidance on the practical usability of the model.

#### 0 5.1 Effect of external load and chemical reactions on solute migration

This subsection investigates how external loading and aqueous reactions jointly influence solute migration and calcite dissolution by varying the magnitude of loading, deformation behaviour, and dissolution-reaction pathways.

Under bottom drainage conditions, all three loading cases demonstrate that a higher load produces a larger initial pore-water pressure, slower dissipation, and greater ultimate settlement (Figures 6a,b). A larger initial excess pressure requires a longer time to dissipate, thus extending the deformation period despite a greater absolute pressure drop. At mid-depth (z = 0.5 m), which is closer to the drainage outlet, the pore pressure dissipates more rapidly and the displacement stabilises earlier. Overall, **Case D** exhibits the highest pore pressure response and settlement, followed by **Case A** and then **Case E**, confirming that increasing load amplifies both pressure evolution and deformation.

Figures 6c—h illustrate the temporal evolution of aqueous concentrations within the calcite-dissolution system under different mechanistic scenarios. In Figure 6c (elemental carbon), **Case B**—which excludes aqueous reactions—yields markedly lower concentrations than the other cases because chemical reactions are suppressed. **Case C**, where matrix deformation is neglected, exhibits a crossover behaviour: its concentration is lower than that of the fully coupled **Case A** at early times but becomes higher at later stages. This reversal arises because deformation during consolidation reduces the pore volume and flow ability, thereby restricting early advective release of dissolved species. In contrast, the absence of deformation preserves pore connectivity and facilitates solute transport, resulting in higher late-time concentrations.

The influence of mechanical compression is further highlighted under varying loading magnitudes (Figure 6d). Initially, higher loads accelerate the release of the solute through increased flow and reduction of porosity, but as compression progresses, the decrease in pore volume and flow ability restrict later stage transport. Consequently, long-term concentrations under heavy loads become lower than those observed under lighter loads.

Unlike Y.C, Y.Ca has no external solute source, the initial calcium concentration is negligible and increases only by calcite dissolution. Accordingly, Figure 6e shows a gradual increase in aqueous Ca<sup>2+</sup> over time, while **Case B** remains essentially zero because no calcium derived from dissolution is supplied. Figures 6e–f further demonstrate that higher loads or the inclusion of deformation compress the soil skeleton, reducing porosity and flow capacity. These mechanical changes alter the transport pathways and influence the rate at which calcium accumulates, underscoring the coupled role of stress, deformation, and chemical reactions in governing long-term redistribution of solutes.

Figures 7 and 8 present two-dimensional contour maps at t = 20 years for dissolved carbon (*Y.C*) originating from calcite dissolution (Fig. 7) and calcium (*Y.Ca*) released by the same process (Fig. 8). Three external source zones are clearly visible, with strengths increasing from left to right. This gradient produces markedly different peak concentrations across the sources—for

Figure 6. Time evolution of pressure, vertical displacement and concentrations under different models (Case A-C, Case D, Case E) at the observation point located at x = 17 m.




instance, under **Case A**, the rightmost source exhibits approximately two to three times higher concentrations than the leftmost one.

In Figure 7, the carbon concentration field follows a similar spatial pattern, reaffirming that neglecting deformation or applying a smaller load accelerates dissolution and enhances solute accumulation. The three distinct source-controlled peaks along the horizontal direction are clearly delineated. Initially, the migration of the solute occurs predominantly in the vertical direction following the main drainage pathway, and with time, it spreads laterally. These maps confirm that the mechanisms identified in the temporal profiles, specifically the effects of deformation, loading magnitude, and reaction coupling, also govern horizontal solute dispersion.

In Fig. 8, the calcium concentration fields exhibit trends consistent with the earlier time series results, now extending into a full spatial perspective. The transition from vertical to horizontal migration highlights preferential transport pathways shaped by loading and deformation. As expected, **Case B** remains near zero because there is no dissolution-reaction source, while the other cases display distinct high concentration regions aligned with the three source zones. Enhanced mechanical compression hinders fluid migration and solute dispersion, leading to narrower plumes and lower concentration peaks than in lighter-load or un-deformed cases such as **Case C** and **Case E**.

**Figure 7.** Contour plots of *Y.C* for the different models (Case A-E).

Figure 9 shows the two-dimensional contour maps of pH at t = 20 years. The pH distribution closely mirrors the dissolved carbon and calcium fields, being primarily governed by local ionic equilibria within the aqueous phase. Zones enriched in dissolved calcium and carbonate species exhibit lower pH (more acidic), while areas with limited accumulation of solutes remain near the background weakly acidic level (pH  $\approx 6.5-7.0$ ). Accordingly, cases or source zones that generate higher solute concentrations develop broader and deeper low pH regions, while non-reactive regions maintain near-neutral conditions.

Figure 10 presents two-dimensional contour maps of calcite dissolution at different times. During the early stage (up to approximately 10 years), dissolution remains concentrated beneath the strongest sources of contaminants. In **Case A**, the

**Figure 8.** Contour plots of *Y.Ca* for the different models (Case A-E).

**Figure 9.** Contour plots of pH for the different models (Case A-E).






upper layer below the rightmost source, where the inflow is strongest, loses nearly 15% of its initial calcite mass by year 10, whereas the depletion beneath the weakest leftmost source remains below 5%.

The results reveal that external loading exerts a pronounced influence on the morphology and evolution of the reactive front. With increasing load, the transition between the reacted and unreacted zones becomes progressively smoother and more diffuse, while in the non-deforming case (**Case C**) it remains narrow and sharply defined. For example, at t = 20 years, the front thickness measured as the horizontal distance of 40% calcite-depletion isolines—widens from approximately 1.4 m in **Case C** to 2 m under high load (**Case A**). Consistently, the two-dimensional maps of aqueous concentration confirm this behaviour: the concentration front is distinctly sharper in the non-deforming case (**Case C**), whereas deformation produces a smoother and more diffuse transition across the reactive zone.

This "mechanical smoothing effect" arises from differences in solute distribution and local chemical disequilibrium. Under higher external load, pore pressure redistribution alters the concentration field, redirecting solute transport predominantly along the horizontal direction rather than vertically. As a result, the dissolved products (e.g.  $Ca^{2+}$ ,  $HCO_3^-$ ) spread laterally and accumulate near the reaction zone, moderating vertical concentration gradients and producing a more diffuse dissolution front. This accumulation increases the local saturation index ( $SI_{calcite}$ ) and reduces the thermodynamic driving force term  $(1-10^{nSI})$ , thus slowing and broadening the dissolution process spatially. Consequently, concentration gradients across the front are gradually smoothed, yielding a wider, more diffuse reactive zone. In contrast, the non-deforming case (**Case C**) maintains stronger undersaturation and steeper concentration gradients, resulting in a sharper, advection-controlled interface that propagates more abruptly through the matrix.

Together, these results demonstrate that dissolution, flow-driven transport, and load-induced pore-space evolution jointly control the timing and magnitude of solute concentrations. The 2D contour maps corroborate the line-based profiles, confirming that these trends persist in the horizontal direction and revealing peak differences and lateral plume structures, such as spreading and source-zone interactions, not captured by 1D analyses.

#### **5.2** Effect of saturation on reactive transport

This subsection examines how saturation conditions influence reactive solute transport in deformable porous media. Figures 11a and b compare excess pore pressure p and vertical displacement  $u_z$  between three representative cases: the near-saturated baseline **Case A** ( $S_r = 0.88$ ), the fully saturated **Case F** ( $S_r = 1.0$ ), and the less-saturated **Case G** ( $S_r = 0.80$ ). The mechanical response strongly depends on the saturation. The fully saturated **Case F** exhibits the highest excess pore pressure—approximately 400 kPa at the top, nearly three times that of **Case G**—because its pores are filled with nearly incompressible water that resists volumetric change under load.

Although all cases reach similar final surface settlements ( $\approx 0.065$  m), **Case F** requires substantially longer to stabilise, as excess pressure dissipates only through slow water drainage in the absence of compressible gas. Lower saturation reduces both the pore water content and the initial pressure peak, facilitating faster dissipation and earlier stabilisation; **Case G**, for instance, shows roughly 45% lower peak p and settles much earlier than **Case F**.

Figure 10. Contour plots of Ys. Calcite for the different models (Case A-E).







Figures 11c–g depict the spatial and temporal evolution of dissolved calcium (*Y.Ca*) and carbon (*Y.C*) at different saturation levels. Both solutes exhibit similar behaviour: lower saturation generally enhances aqueous concentrations. Comparison in the three case groups (11e–g) clarifies the distinct and combined functions of deformation, saturation, and geochemical reactions.

The Case B, J, and K group disables geochemical reactions to isolate the influence of mechanical deformation. Although total compression is nearly identical in all cases, the drainage rate varies markedly with saturation. In fully saturated condition, excess pore pressure slowly dissipates and the resulting limited fluid mobility restricts advective transport, leading to lower solute concentrations. In contrast, partially saturated conditions allow faster drainage and stronger advection, which promote vertical redistribution. Thus, the observed concentration difference primarily reflects drainage-controlled advection capacity rather than differences in overall deformation magnitude.

In contrast, the **Case C, H, I** group suppresses deformation but retains geochemical reactions, thus isolating the reaction-driven response to varying saturation. Here, decreasing  $S_r$  elevates the concentrations mainly because the available water volume ( $\theta_w = nS_r$ ) diminishes; thus, the same mass of the dissolution-derived solute is distributed into a smaller liquid volume, producing uniformly higher aqueous concentrations. This  $1/\theta_w$  scaling amplifies the contrast between saturation levels.

Finally, the **Case A, F, G** group activates both deformation and geochemical reactions, and the observed patterns reflect their combined influence. Geochemical processes increase the levels of solutes through ongoing dissolution, while lower saturation further magnifies these increases by reducing the aqueous volume. Meanwhile, mechanical compression partially counteracts this enhancement by reducing the level of pore connectivity and limiting the degree of advective redistribution. In general, geochemical reactions amplify saturation-driven contrasts, whereas mechanical deformation provides a modest damping effect on solute concentrations under any given saturation.

Figures 12a–i present the two-dimensional distributions of dissolved carbon (Y.C) at t = 20 years for all case groups. Compared with earlier vertical profiles, these maps emphasise the pronounced horizontal variability shaped by the three contaminant sources. Peak concentrations consistently occur near the rightmost source where the inflow is strongest, reaching  $Y.C \approx 3.0 \text{ mol/L}$ , approximately two to three times higher than those near the weakest leftmost source. Horizontal plumes are progressively broadened from right to left, revealing distinct source–plume separation and limited mixing between adjacent zones.

Lower saturation elevates aqueous concentrations throughout the domain, particularly around the strong right-hand source, where the high-solute cores expand laterally by about 15–20% in **Case G** relative to **Case F**. Similar conclusions can be drawn from **Case J** and **Case K**. In contrast, mechanical deformation exerts an opposing influence: at the same saturation level, the peak *Y.C* values in deformable cases (e.g., **Case G** versus **I**) are about 20% lower, and the corresponding high-concentration plumes appear more confined and compact.

The pH distributions (Figures 12j–o) reflect the coupled influence of dissolved calcium, carbonate species, and other ions on local aqueous equilibria. Regions with higher solute concentrations develop broader and more acidic zones, while areas with limited solute accumulation remain close to neutral. Consequently, the extent of acidification closely mirrors both the intensity and lateral extent of the solute plumes.

Figure 11. Time evolution of pressure, vertical displacement and concentrations under different models (Case A-C, Case F-K) at the observation point located at  $x = 17 \,\mathrm{m}$  (to be continued).

Overall, these results confirm that the solute and acidity peaks spatially align with the sources of the contaminant. Lower saturation amplifies both the magnitude and the horizontal reach of high-concentration plumes, while mechanical deformation slightly suppresses the concentration peaks and contracts the plume footprints. Together, these trends highlight the competing effects of saturation and deformation on the long-term spread of reactive solutes in deformable porous media.

# 5.3 An example of multi-mineral reaction



The purpose of this subsection is to demonstrate the capability of the developed solver to handle multi-component and multi-mineral reactions, thus establishing a foundational framework for future model extensions.

Figures 13 and 14 illustrate the temporal evolution of aqueous chemistry and mineral reactions driven by three externally supplied contaminants, carbon (Y.C), iron (Y.Fe), and aluminium (Y.Al), with source intensities decreasing from left to right. Among the three, the external influx of iron is the strongest, followed by aluminium, while carbon has the weakest source. However, the retained aqueous concentrations show the opposite order (C > Fe > Al) due to their distinct reaction pathways. Dissolved carbon maintains the highest residual level, with typical peaks around 6.6 mol/L, while dissolved iron and aluminium



Figure 11. Time evolution of pressure, vertical displacement and concentrations under different models (Case A-C, Case F-K) at the observation point located at x = 17 m.

reach lower quasi-steady concentrations of approximately 0.2 mol/L and 0.12 mol/L, respectively. Aluminium undergoes the most intensive reaction: the introduced  $Al^{3+}$  rapidly precipitates as *gibbsite* ( $Al(OH)_3$ ), leaving only trace amounts of dissolved aluminium in the aqueous phase. As a result, the aluminium plume appears nearly stationary, giving the impression of limited solute migration despite strong local reactions.

Initially, the injected source solution is mildly alkaline due to the presence of hydroxyl- and carbonate-bearing complexes such as Al(OH)<sub>4</sub> and HCO<sub>3</sub>. As reactions proceed, the precipitation of gibbsite (Al(OH)<sub>3</sub>) and siderite (FeCO<sub>3</sub>) releases protons through hydrolysis and carbonate exchange, progressively neutralising the alkalinity and driving the system towards mildly acidic conditions. Calcite (CaCO<sub>3</sub>) also precipitates, but plays a dual role by partially buffering the acidity generated through its dissolution equilibrium. Spatially, the right-hand side exhibits a faster pH decline than the left-hand side, primarily because of stronger precipitation reactions and weaker carbonate buffering in that region.

Calcite remains persistently oversaturated throughout the simulation, indicating a continuous but slow precipitation. This sluggish behaviour arises from two limiting factors: low concentration  $Ca^{2+}$  and a mildly acidic environment (pH  $\approx$  6.0). Under such conditions, most dissolved inorganic carbon (DIC) exists as  $HCO_3^-$  rather than  $CO_3^{2-}$ , thus reducing the effective carbonate

**Figure 12.** Contour plots of *Y.C* and *pH* for the different models (Case A-C, Case F-K).

Figure 13. Contour plots of aqueous concentrations and pH for Case L.






activity required for calcite growth. Consequently, net calcite precipitation proceeds slowly, achieving only a modest increase of about 15% of its initial mass by year 50. Although the process consumes small amounts of Ca<sup>2+</sup> and bicarbonate ions, the resulting buffering capacity is insufficient to neutralise the protons released from metal hydrolysis and gibbsite formation. Hence, calcite acts as a slow but persistent secondary precipitate phase under calcium- and pH-limited conditions.

Siderite forms at a moderate but sustained rate where injected  $Fe^{2+}$  overlaps with carbonate species supplied by the carbon source (Y.C, primarily  $HCO_3^-/CO_3^{2-}$ ). Concomitant precipitation of calcite competes for carbonate and thus moderates siderite growth. Compared with gibbsite, the siderite front advances more slowly and extends over a broader region. By year 30, siderite captures roughly 25–30% of the incoming iron along the primary flow paths, increasing to approximately 40–45% by year 50. As a result, dissolved iron concentrations peak near the source, governed by the source strength, and gradually attenuate downstream under reaction control.

Upon arrival of *Y.AI*, gibbsite precipitates rapidly, causing an abrupt decrease in the dissolved aluminium near the source. Quantitatively, more than 80% of the incoming Al is removed within the first 10 years in the proximal zone, and the cumulative fraction increases from approximately 5% initially to nearly 20% by year 50. Because the precipitation rate greatly exceeds the advective transport rate, most reactions remain confined to the source region, leaving negligible aluminium migration downstream. This behaviour characterises a strongly reaction-controlled regime, where rapid local precipitation suppresses long-range solute propagation.

In general, the solver consistently reproduces these coupled processes: progressive plume expansion over 10–50 years, clear dependence on source strength, moderated pH evolution, and mass-conserving behaviour across all solutes. These results confirm that the model robustly simulates complex, multi-mineral reactive transport involving concurrent precipitation and dissolution fronts.

## 5.4 Computational performance

All simulations were performed on a workstation equipped with an **Intel Xeon Silver 4114 CPU** (2.20 GHz) and 256 GB **RAM**. Unless otherwise specified, each case was solved using **four parallel MPI processes** under the same hardware configuration to ensure consistency and fair comparison of CPU time and memory usage.

To evaluate numerical efficiency and assess the trade-off between computational cost and accuracy, this subsection compares the performance of the full HMC solver with its simplified counterparts, in which either the chemical module is deactivated. The evaluation considers four key indicators: total CPU time, average time per step, maximum RAM usage, and the relative error of solute concentration at the most responsive monitoring point. For each paired comparison, the fully coupled HMC case under identical loading and saturation conditions (Cases A, F, G) serves as the reference, while the simplified (B, K) and multi-mineral (L) cases are evaluated against it.

The relative error ( $RE_Q(\%)$ ) quantifies the deviation of a simplified model from its corresponding full HMC reference under the same loading and saturation. For a time-dependent variable Q(t) (e.g., carbon concentration Y.C) measured at a monitoring point,  $RE_Q(\%)$  is defined using the  $L^2$ -norm of the temporal series, normalised by the reference:


Figure 14. Contour plots of mineral concentrations for Case L.

615 
$$\operatorname{RE}_{Q}(\%) = \frac{\left\| Q_{\operatorname{model}}(t) - Q_{\operatorname{HMC,ref}}(t) \right\|_{L^{2}(0,T)}}{\left\| Q_{\operatorname{HMC,ref}}(t) \right\|_{L^{2}(0,T)}} \times 100,$$
 (21)

where  $||X||_{L^2(0,T)} = \sqrt{\int_0^T X^2(t) dt}$ . In this study,  $RE_Q(\%)$  is evaluated at the most responsive monitoring point identified from the HMC baseline for each comparison.

Table 5 summarises the computational cost and relative error evaluated at the top monitoring point (x = 17 m, z = 1 m). All reported performance metrics correspond to the output of the first MPI process (rank 0). Complete HMC simulations (**A**, **F**, **G**) required approximately 25–26 h of CPU time, while deactivation of chemical reactions (**B**, **J**, **K**) reduced total runtime by more than two orders of magnitude, with almost identical memory consumption. The nearly constant use of RAM arises because all cases share the same mesh and field allocations; enabling a module does not significantly alter the solver's data structure or storage requirements. However, this simplification introduces substantial concentration deviations (RE<sub>c</sub>  $\approx$  44%), demonstrating the importance of retaining chemical coupling to capture realistic solute behaviour. In contrast, the multi-mineral reactive case (**L**) exhibited the highest computational demand, reflecting the increased cost associated with solving the extended reaction

network. Overall, these results confirm that while simplified configurations offer significant computational savings, they do so at the expense of accuracy, underscoring the necessity of HMC modelling for quantitative predictive analyses.

**Table 5.** Computational cost and relative error at the top responsive point ( $x = 17 \,\text{m}, z = 1 \,\text{m}$ ).

| Pair                                    | Model             | Total CPU (h)  | Avg/step (s) | RAM (MB) | $RE_{Y.C}$ (%) |
|-----------------------------------------|-------------------|----------------|--------------|----------|----------------|
| $\mathbf{A} \leftrightarrow \mathbf{B}$ | A (HMC ref.)      | 24.96          | 44.91        | 66.75    | _              |
|                                         | B (no-chem)       | 0.16           | 0.29         | 66.84    | 44.17          |
| $F \leftrightarrow J$                   | F (HMC ref.)      | 26.20          | 47.14        | 66.70    | _              |
|                                         | J (no-chem)       | 0.23           | 0.41         | 67.03    | 44.16          |
| $G \leftrightarrow K$                   | G (HMC ref.)      | 26.14          | 47.02        | 66.78    | _              |
|                                         | K (no-chem)       | 0.17           | 0.31         | 67.06    | 44.22          |
| $A \leftrightarrow L$                   | A (HMC ref.)      | 24.96          | 44.91        | 66.75    | _              |
|                                         | L (multi-mineral) | 32.38 (20 yrs) | 58.26        | 67.22    | _              |

Notes: CPU times measured on the same hardware; RAM = maximum resident set size; "—" = not applicable.

# 6 Conclusions

In this study, a reactive transport modelling framework is proposed by embedding a flexible geochemical module into an existing hydro-mechanical solver, enabling consistent treatment of multi-component aqueous species and multiple mineral phases in deformable porous media.

The results show that calcite dissolution, flow-driven transport, and load-induced pore-structure evolution jointly govern both the timing and spatial distribution of solutes. Mechanical compression accelerates early-stage solute release by enhancing advective flow but later suppresses transport as porosity and flow ability decline, leading to non-monotonic concentration trends. Neglecting deformation or applying a smaller external load promotes faster mineral dissolution, whereas deformation under high load suppresses the reaction and yields a smoother dissolution front.

Saturation strongly influences both the hydro-mechanical response and the reactive-solute distribution. Fully saturated soils develop the highest excess pore pressures and dissipate them most slowly, whereas lower saturation lowers the pressure peak and accelerates settlement. For solutes, decreasing saturation consistently elevates aqueous concentrations by increasing the flow velocity and concentrating dissolution-released species; this amplification is most pronounced near strong contaminant sources, which expands the high-solute and low-pH plumes laterally. Mechanical deformation counteracts this effect by reducing connected pore volume and advective transport, slightly damping concentrations, and confining plume spread.

However, the present study lacked laboratory or field data for quantitative validation and did not yet consider feedbacks of geochemical reactions on key hydraulic-mechanical properties such as permeability and porosity. Future work will incorporate reaction-induced property changes and benchmark the solver against experimental datasets to improve its predictive reliability.

# Appendix A: Proportion of elements diffusion coefficient

Within the coupled hydro-mechanical-chemical framework, the relative fractions of aqueous species  $\omega_i$  associated with each element can be dynamically obtained or prescribed as constants. When executed in a fully coupled mode with PhreeqcRM, the speciation module automatically determines species concentrations in each cell and time step, allowing  $\omega_i = Y_i/Y_{\text{element}}$  to evolve with local chemistry. This allows the effective diffusion coefficient  $D_m = \sum_i \omega_i D_m^{(i)}$  to vary consistently with reaction, pH, and ionic strength. The species-specific diffusion coefficients  $D_m^{(i)}$  are retrieved directly from the Phreeqc database via the -diffusion\_coefficient entries, ensuring a thermodynamically consistent parameterisation for both static and dynamic simulations.

In this study, the fractions  $\omega_i$  are prescribed rather than dynamically updated, as shown in Figures A1 and A2. These figures present the compositions, fractions, and molecular diffusion coefficients of predefined species adopted in the model test cases. The values are fixed input parameters for clarity and reproducibility, but can be easily modified to represent alternative systems without changing the numerical framework. The present work primarily aims to capture the coupled deformation, solute migration, and geochemical reactions under near-saturated conditions, while dynamic updates of species fractions can be incorporated in future extensions of the same framework.

Figure A1. The proportion of elements diffusion coefficient for single mineral reaction.

## 660 Appendix B: Implementation of the geochemical-window controller

The geochemical-window controller (ChemWindow class) manages when and how the chemical module is activated during the global time-marching loop of the hydro-mechanical-chemical solver. It parses user-defined activation periods from an external plain-text file and supplies the transport solver with the corresponding sub-stepping parameters whenever chemistry is active.

Figure A2. The proportion of elements diffusion coefficient for multi-mineral reaction.

Each entry in the configuration file is defined as a quadruple:

# 665 startStep nSteps $\Delta t_{\mathrm{chem}}$ nSub

670

680

where startStep specifies the global transport-step index at which activation begins; nSteps is the duration of the window;  $\Delta t_{\text{chem}}$  is the reaction sub-time-step size; and nSub is the number of reaction sub-steps within each transport step. **Note:** both startStep and nSteps refer to *time index* maintained by OpenFOAM's time-loop manager rather than literal integer step numbers; the time index advances according to the internal time control of the solver and may not be uniformly incremental.

Upon initialisation, the class reads all entries, removes comments and non-printable characters, resolves overlaps using a selected *conflict-resolution policy* (e.g. *LastWins*, *FirstWins*, *MinDt*, *MaxDt*), and merges adjacent ranges that share identical sub-stepping parameters. The result is a sorted, non-overlapping list of ChemRange objects that can be queried efficiently at runtime.

During each global transport step, the solver calls

#### 675 inChemWindow(currentStep, newDt, nSub)

to determine whether the chemical module is active. If the current step falls inside a defined window, the routine returns the sub-time-step size (newDt) and the required number of reaction sub-steps (nSub); otherwise, the solver advances only the hydro-mechanical and transport modules. This design avoids unnecessary chemical calculations outside the specified windows, thereby reducing computational overhead.

# A simplified pseudo-code of the driver loop is:

```
1: initFromFile("chemWindow.txt"); // parse and build ranges
2: for (step = 0; step 

Author contributions. Bolin Wang: Conceptuation, Methodology, Data curation, Visualization, investigation, validation, writing-original draft; Dong-Sheng Jeng: Conceptuation, Methodology, Writing-review & editing, supervision, Resources, funding acquisition, project administration.

Competing interests. The authors confirm that there is no competing interest are present

Acknowledgements. The authors are grateful for the support from Shandong Provincial Overseas High-Level Talent Workstation (A2021-700 140).

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
