# Peer review of "Multi-component reactive transport in near-saturated deformable porous media"

_EGUsphere, 2025_

## Author Comment (AC1)

**Response Statement to Community's Comments (Giacomo Medici)**

**Wang and Jeng**

**January 24, 2026**

The authors thank the reviewer for the valuable comments. The manuscript has been revised by carefully considering all the comments. The changes are highlighted in the marked copy, and detailed responses to the reviewer's comments are provided below.

**General Comment:**

*Good research on contaminant transport in porous media that needs some improvement and further detail. See my specific comments, they will improve the potential new version of the manuscript.*

**Response (General Comment):**

We thank the reviewer for the positive assessment of our work and for the constructive comments. All specific suggestions have been carefully considered and addressed in the revised manuscript, leading to improvements in clarity, technical detail, and overall presentation.

**Comment #1-1:**

*Line 23. . . ."has been the most widely applied framework in the field". Last words not backed up by references. Add these papers that incorporate discussion on evidence of the Fick's law in the field:*

*- Agbotui, P.Y., Firouzbehi, F., Medici G. 2025. Review of effective porosity in sandstone aquifers: insights for representation of contaminant transport. Sustainability 17, no. 14 (2025): 6469.*

*- Parker, B.L., Cherry, J.A., Wanner, P. 2022. Determining effective diffusion coefficients of chlorohydrocarbons in natural clays: unique results from highly resolved controlled release field experiments. Journal of Contaminant Hydrology 250, 104075.*

**Response:**

We thank the reviewer for this insightful comment. To address this concern, we have revised the statement to avoid an overly strong formulation and have clarified the applicability of the advection–dispersion equation by explicitly referencing field-scale evidence supporting Fickian diffusion in natural geological media. In particular, we have added recent field-based studies by Parker et al. (2022) and Agbotui et al. (2025) to support the revised statement.

[Deleted content:]

[Added new content:] *remains one of the most widely applied frameworks for modelling contaminant transport in porous media, with its relevance supported by field-scale evidence of Fickian diffusion in natural geological formations (Parker et al., 2022; Agbotui et al., 2025).*

**[Line 23–25]**

**Comment #1-2:**

*Line 30. This sentence does not work as it is. Before "purely physical transport" and then you involve geochemistry. Please, revise the structure.*

**Response:**

We thank the reviewer for pointing out this issue. The sentence has been restructured to clearly distinguish between traditional contaminant transport formulations that focus on purely physical processes and the additional role of geochemical reactions. This revision resolves the logical inconsistency and improves the clarity of the statement.

[Deleted content:]

[Added new content:] *While many contaminant transport models focus primarily on purely physical processes, geochemical reactions can also significantly influence the migration of dissolved species by altering concentration distributions and species mobility.*

**[Line 31–34]**

**Comment #1-3:**

*Line 76. You need to disclose the general aim of the research at the end of the introduction.*

**Response:**

We thank the reviewer for this comment. A clear statement of the general aim of the study has now been added at the end of the Introduction.

[Added new content:] *The general aim of this study is to develop a three-dimensional hydro-mechanical–chemical (HMC) modelling framework that explicitly couples hydro-mechanical deformation with multicomponent geochemical reactive transport. The framework is used to investigate solute transport in near-saturated deformable porous media. Specifically, the objectives of this study are to: (i) analyse the coupled effects of mechanical deformation and multicomponent geochemical reactions on solute transport within the proposed HMC framework; (ii) explore the sensitivity of deformation-driven solute transport to loading conditions and limited deviations from full saturation through numerical simulations.*

**[Line 75–76]**

**Comment #1-4:**

*Line 76. You need to describe the specific objectives of your research by using numbers (e.g., i, ii, and iii) at the end of the introduction.*

**Response:**

We thank the reviewer for this comment. The specific objectives of the study have now been explicitly stated at the end of the Introduction using numbered points (i–iii).

[Added new content:] *Specifically, the objectives of this study are to: (i) represent deformation induced by mechanical loading under near-saturated conditions within a three-dimensional HMC framework and quantify its influence on solute transport; (ii) incorporate multicomponent and multi-mineral geochemical reactions into the HMC framework and examine their role in deformation-driven solute migration; (iii) investigate the effects of loading conditions and degrees of saturation on solute transport behaviour through numerical simulations.*

**[Line 76–80]**

**Comment #1-5:**

*Line 199. Insert reference to Parkhurst for PHREEQC.*

**Response:**

We thank the reviewer for this comment. In addition to the existing reference for PhreeqcRM, the standard reference for the PHREEQC geochemical engine by Parkhurst and Appelo has now been explicitly added in the

revised manuscript.

[Added new content:] which preserves the full thermodynamic and kinetic reaction capabilities of PHREEQC (Parkhurst and Appelo, 2013) while streamlining data exchange between the reactive-transport code and the geochemical engine.

**[Line 208–209]**

**Comment #1-6:**

*Line 250. This one looks to me a figure, not a table or an equation. Am I correct?*

**Response:**

We thank the reviewer for this comment. The item at Line 250 has now been presented and labelled as a figure, with the corresponding figure caption and in-text references revised accordingly.

[Added new content:]

[Figure]

Figure 1: Schematic of reaction–transport time stepping with and without a chemistry window.

**[Line 258–259]**

**Comment #1-7:**

*Line 700. More detail on the acknowledgement. Funding bodies are unclear.*

**Response:**

We thank the reviewer for this comment. The Acknowledgements section has been revised to clarify the funding body and the nature of the support provided.

[Deleted content:]

[Added new content:] *The authors gratefully acknowledge the institutional support from the Shandong Provincial Overseas High-Level Talent Workstation (Grant No. A2021-140) for facilitating the research presented in this paper.*

**[Line 707–709]**

**Comment #1-8:**

*Figure 3. Make the image larger.*

**Response:**

Thank you for the suggestion. The size of Figure 3 has been increased to improve readability.

[Added new content:]

[Figure]

Figure 2: Comparison of the present model (solid line) with Benchmark 2 (Pavuluri et al. (2022), diamond markers). The blue line represents the result at 20 minutes, the red line at 40 minutes, and the black line at 60 minutes.

**[Line 325–326]**

**Comment #1-9:**

*Figure 5. Same here, make the figure larger.*

**Response:**

Thank you for the suggestion. The size of Figure 5 has been increased to improve readability.

[Added new content:]

[Figure]

Figure 3: Load pattern under different models: the black line represents **Case A**, the red line represents **Case D**, and the blue line represents **Case E**.

**[Line 421–422]**

**Comment #1-10:**

*Two figures 11. This is not ok.*

**Response:**

Thank you for pointing this out. The figure was originally split across two pages because the composite figure was too large to fit on a single page. In the revised manuscript, this issue has been resolved by separating the content into two figures: one showing pore pressure and vertical displacement, and the other showing concentration evolution.

[Added new content:]

[Figure]

(a) $p$ vs $t$ of **Case A, F, G**

(b) $u_z$ vs $t$ of **Case A, F, G**

Figure 4: Time evolution of pressure and vertical displacement under different models (**Case A-C**, **Case F-K**) at the observation point located at $x = 17$ m.

[Figure]

(a) $Y.Ca$ vs $t$ of **Case A, F, G**

(b) $Y.Ca$ vs $t$ of **Case C, H, I**

(c) $Y.C$ vs $t$ of **Case A, F, G**

(d) $Y.C$ vs $t$ of **Case C, H, I**

(e) $Y.C$ vs $t$ of **Case B, J, K**

Figure 5: Time evolution of concentrations under different models (**Case A-C**, **Case F-K**) at the observation point located at $x = 17$ m.

**Comment #1-11:**

*Figure 9. Make the figure larger also here.*

**Response:**

Thank you for the suggestion. The size of Figure 9 has been increased to improve readability.

[Added new content:]

[Figure]

Figure 6: Contour plots of *pH* for the different models (**Case A-E**).

[Line 438–447]

**References**

Agbotui, P.Y., Firouzbehi, F., Medici, G., 2025. Review of effective porosity in sandstone aquifers: insights for representation of contaminant transport. Sustainability 17, 6469. doi:https://doi.org/10.3390/su17146469.

Parker, B.L., Cherry, J.A., Wanner, P., 2022. Determining effective diffusion coefficients of chlorohydrocarbons in natural clays: unique results from highly resolved controlled release field experiments. Journal of Contaminant Hydrology 250, 104075. doi:https://doi.org/10.1016/j.jconhyd.2022.104075.

Parkhurst, D.L., Appelo, C.A.J., 2013. Description of Input and Examples for PHREEQC Version 3: A Computer Program for Speciation, Batch-Reaction, One-Dimensional Transport, and Inverse Geochemical Calculations. Technical Report Techniques and Methods, Book 6, Chapter A43. U.S. Geological Survey. Reston, VA, USA.

Pavuluri, S., Tournassat, C., Claret, F., Soulaine, C., 2022. Reactive transport modeling with a coupled openfoam®-phreeqc platform. Transport in Porous Media 145, 475–504. doi:https://doi.org/10.1007/s11242-022-01860-x.

---

## Author Comment (AC2)

**Response Statement to Community's Comments (CC3)**

Wang and Jeng

January 24, 2026

The authors thank the reviewer for the valuable comments. The manuscript has been revised by carefully considering all the comments. The changes are highlighted in the marked copy, and detailed responses to the reviewer's comments are provided below.

**Comment #CC3:**

*You have two verification benchmarks, but the paper itself acknowledges no lab/field quantitative validation. While this isnt a fatal flaw, you do need to strengthen credibility with additional numerical evidence. Consider adding (1) Grid/time-step sensitivity for one representative case (even a coarse/medium/fine study + one plot at the monitoring point). (2) Splitting / sub-stepping sensitivity: show that results don't materially change when the chemistry sub-step size changes within a window (or quantify the trade-off). Your own text highlights synchronisation issues, so you should demonstrate control. (3) For benchmark 2, quantify mismatch: e.g., L2 error vs time, and show it decreases with refinement or explain the irreducible discrepancy (boundary formulation differences are mentioned but not demonstrated).*

**Response:**

We thank the reviewer for this constructive comment and for the specific suggestions on strengthening the numerical credibility of the framework in the absence of laboratory or field-scale validation.

In response, we have augmented the manuscript with additional numerical evidence addressing points (1)–(3) as follows. First, a grid and time-step sensitivity analysis has been added for a representative test case, using coarse, medium, and fine discretisations. The results are evaluated at a monitoring location and demonstrate that the key hydro-mechanical and chemical responses are insensitive to further refinement within the investigated resolution range.

Second, we have extended the splitting and chemistry sub-stepping sensitivity analysis to explicitly demonstrate that the results remain stable when the chemical sub-step size is varied within a prescribed window. This

directly addresses the synchronisation and operator-splitting considerations highlighted in the manuscript and confirms that the adopted coupling strategy is numerically controlled.

Third, for Benchmark #2, the discrepancy is dominated by differences in the advective transport formulation: the benchmark employs a conservative face-based volumetric flux, whereas the present framework represents advection through the divergence of a Darcy-based, cell-centred flux under near-saturated conditions. These formulations are not equivalent at the discrete level, and therefore a non-zero irreducible mismatch remains even under mesh and time-step refinement.

Taken together, the added sensitivity analyses and quantitative error assessment demonstrate that the present implementation is numerically stable, well controlled with respect to discretisation and splitting choices, and that the observed benchmark discrepancy reflects structural differences between the governing formulations rather than numerical inconsistency.

[Deleted content:] ~~The simulated concentration profiles of dissolved Ca were compared with those of the reference study, as illustrated in Figure. These two solutions agree well at intermediate and late times, but noticeable deviations appear in the early stage of the breakthrough. These differences arise primarily from variations in the underlying transport formulations and boundary representations. The present solver accounts for near-saturation effects, which modify the advective flux through compressibility terms, whereas the reference benchmark adopted a classical ADE-based formulation with constant porosity and a prescribed uniform flow field. As the dissolution front advanced and concentration gradients became less steep, the two solutions gradually converged, although a small residual discrepancy persisted because of the fundamentally different governing equations and kinetic parameterisations.~~

[Added new content:] *Within the OpenFOAM framework adopted in this study, the governing equations were discretised using the finite volume method. Diffusive and dispersive fluxes were evaluated using the Gauss linear scheme, while the advective term of solute transport was discretised using a Gauss limitedLinear scheme with a limiter coefficient of 1 in order to suppress spurious oscillations. Temporal integration was performed using the implicit Euler method. The pressure, displacement and concentration equations were solved using the GAMG solver with a DILU preconditioner, with absolute and relative tolerances of $10^{-9}$ and zero, respectively. The coupled HMC system was advanced using a segregated outer-iteration strategy, and convergence was achieved when the residuals of all primary variables dropped below $10^{-6}$. Further implementation details are provided in the companion paper (Wang and Jeng, 2025).*

*To strengthen the numerical credibility of the proposed HMC framework, a systematic sensitivity analysis was conducted with respect to spatial discretisation, global time-step size and chemical sub-stepping in the ChemWindow controller. A one-at-a-time strategy was adopted, whereby only one numerical factor was varied while all others were kept identical.*

The numerical deviations were quantified using the mean relative error (MRE) and maximum relative error (MaxRE), defined as

$$MRE = \frac{1}{N} \sum_{i=1}^{N} \left| \frac{y_i - y_i^{\text{ref}}}{y_i^{\text{ref}}} \right|, \quad MaxRE = \max_{1 \le i \le N} \left| \frac{y_i - y_i^{\text{ref}}}{y_i^{\text{ref}}} \right|, \tag{1}$$

where $y_i$ denotes the computed value at the $i$-th comparison point and $y_i^{\text{ref}}$ is the corresponding value obtained from the finest reference solution. The MRE reflects the overall deviation level, whereas the MaxRE captures the largest local discrepancy associated with potential non-linear or synchronisation effects.

The reference configuration corresponds to the finest grid ($120 \times 1 \times 100$), the smallest global time step ($\Delta t = 1.58 \times 10^5$ s), and the smallest chemical sub-step ($\Delta t_c = 3000$ s). The medium discretisation employs a grid of $60 \times 1 \times 50$ with $\Delta t = 3.15 \times 10^5$ s and $\Delta t_c = 6000$ s, while the coarse setting uses $30 \times 1 \times 25$, $\Delta t = 6.31 \times 10^5$ s, and $\Delta t_c = 12000$ s.

Table 1: Grid-, time-step-, and chemical-substep-independence summary.

| Variable | Grid mean (%) | Grid max (%) | Time-step mean (%) | Time-step max (%) | Chem. sub-step mean (%) | Chem. sub-step max (%) |
|---|---|---|---|---|---|---|
| $p$ | $1.2 \to 0.43$ | $1.9 \to 0.45$ | $0.25 \to 0.08$ | $0.27 \to 0.09$ | – | – |
| $\mathbf{u}_s$ | $0.54 \to 0.07$ | $0.92 \to 0.11$ | $0.76 \to 0.25$ | $0.77 \to 0.26$ | – | – |
| Y.C | $5.5 \to 2.1$ | $11.5 \to 3.8$ | $0.15 \to 0.05$ | $0.35 \to 0.11$ | $0.7 \to 0.4$ | $1.1 \to 0.6$ |

Table 1 together with Figs. 1 and 2 summarises the grid-, time-step- and chemical-substep-independence results for pore pressure $p$, solid displacement $\mathbf{u}_s$ and solute concentration Y.C. For the hydro-mechanical variables $p$ and $\mathbf{u}_s$, both the MRE and MaxRE decrease rapidly with mesh refinement and global time-step reduction. In all medium–fine comparisons, the maximum relative errors remain below 0.5%, indicating excellent numerical convergence of the hydro-mechanical part of the solver.

The solute concentration Y.C exhibits a higher sensitivity to spatial resolution, as expected for advection–dispersion–reaction dominated processes. Nevertheless, the MaxRE decreases from 11.5% in the coarse–fine comparison to 3.8% in the medium–fine comparison, while the MRE reduces from 5.5% to 2.1%, demonstrating satisfactory convergence of the transport component. In contrast, the sensitivity to the global time-step size is negligible, with maximum discrepancies below 0.2%. More importantly, decreasing the chemical sub-step size from 12,000 s to 3,000 s results in less than 1% variation in solute concentration, with the MaxRE reducing from 1.1% to 0.6%. This confirms that the proposed ChemWindow synchronisation scheme is numerically stable and does not introduce artificial splitting errors.

Based on these results, the medium discretisation settings ($60 \times 1 \times 50$, $\Delta t = 3.15 \times 10^5$ s, $\Delta t_c = 6000$ s) are adopted in the remainder of this study as a balanced compromise between numerical accuracy and computational

*efficiency.*

[Figure]

(a) $p$ under varying grids    (b) $u_z$ under varying grids    (c) $Y.C$ under varying grids

(d) $p$ under varying time steps    (e) $u_z$ under varying time steps    (f) $Y.C$ under varying time steps

Figure 1: Analysis of grid and time step independence: Fine grid and small time step (round), medium grid and step (square), coarse grid and large step (triangle).

[Figure]

Figure 2: Analysis of chemical-substep independence: small substep (round), medium substep (square), large substep (triangle).

**[Line 323–339]**

*The simulated concentration profiles of dissolved Ca were compared with those reported in the reference study, as shown in Fig. Overall, the two solutions exhibit good agreement at intermediate and late times, whereas noticeable discrepancies arise during the early breakthrough stage.*

*The discrepancies observed between the two solutions are primarily attributed to the fundamentally different discrete treatments of the advective transport term, which lead to distinct boundary flux reconstructions. The*

*reference solver evaluates the advective operator using the surface-based volumetric flux field $\phi$ in the form $\nabla\cdot(\phi Y_i)$, whereas the present framework formulates advection in terms of the divergence of a Darcy-based, cell-centred mass flux, $\nabla\cdot(\mathbf{v_f} Y_i)$. Owing to the distinct surface-flux and volume-flux formulations, the two operators are not discretely equivalent.*

*Consequently, even when identical fixedValue boundary conditions are prescribed, the resulting advective fluxes differ in the discrete implementation, because $\phi$ is imposed directly at cell faces, while $\mathbf{v_f}$ is defined at cell centres and subsequently interpolated to the boundary faces. This structural difference cannot be expected to vanish systematically through mesh refinement and therefore gives rise to irreducible discrepancies during the early transient stage.*

*Within this context, Benchmark #2 is not intended to provide strict equation-to-equation validation against the reference solution, but rather to assess the correctness and feasibility of the proposed governing equations under comparable conditions. The observed agreement in the overall trend and magnitude of the concentration profiles supports the correctness of the present implementation and indicates that the proposed control equations capture the dominant dissolution dynamics. The remaining differences are mainly attributable to the distinct discrete treatments of the advective operator.*

**[Line 323–339]**

**References**

Wang, B.L., Jeng, D.S., 2025. Three-dimensional model for consolidation-induced solute transport in a nearly saturated porous medium. International Journal for Numerical and Analytical Methods in Geomechanics 49, 4436–4464. doi:https://doi.org/10.1002/nag.70070.

---

## Author Comment (AC3)

**Response Statement to Community's Comments (CC2)**

**Wang and Jeng**

**January 24, 2026**

The authors thank the reviewer for the valuable comments. The manuscript has been revised by carefully considering all the comments. The changes are highlighted in the marked copy, and detailed responses to the reviewer's comments are provided below.

**Comment #CC2:**

*Right now, near-saturated risks sounding like a convenience assumption rather than a controlled approximation. You explore Sr 0.80–1.0, but (as written) saturation appears prescribed rather than evolving, which is a big modeling decision for landfill liners and many barrier systems. Please add a short model scope paragraph early (end of Introduction or start of Methods) stating explicitly: (1) saturation is constant in time (if that's the case), (2) when this is reasonable (e.g., quasi-saturated low gas mobility, slow infiltration variability relative to consolidation time, etc.), (3) when it is not (drying fronts, strong unsaturated flow, gas phase transport). Consider adding a brief comparison to Richards/unsaturated THMC literature (even if you don't implement it). And be explicit whether near-saturated here means "two-phase but simplified compressibility" vs "single-phase with effective compressibility."*

**Response:**

We thank the reviewer for this important comment. A dedicated model-scope paragraph has been added at the assumptions to clarify that the degree of saturation is prescribed as a constant and that the formulation corresponds to a single-phase liquid flow framework under near-saturated conditions, with residual gas treated as immobile. The paragraph also specifies the physical regimes in which this approximation is appropriate (e.g. quasi-saturated, low-permeability geomaterials with slow saturation variations) and those in which it becomes invalid (e.g. drying fronts, strong transient infiltration or connected gas pathways). A brief comparison with unsaturated multi-field coupling frameworks is additionally provided to delineate the scope of the present study.

[Deleted content:]

~~continuous network that governs flow, while the gas phase is treated as isolated, immobile bubbles that do not participate in advection. The degree of saturation is therefore prescribed as a constant rather than solved as a dynamic variable, representing conditions where the liquid phase dominates and no significant air invasion occurs. Under this assumption, capillary effects, gas-phase flow, and the transport of dissolved or volatile elements through the gas phase are excluded from the model. The constant saturation also implies that its potential influence on the reaction surface area or transport coefficients is neglected. This simplification is appropriate for wet, deformable porous media in which the liquid phase remains continuous and the gas phase persists only as immobile bubbles; it becomes invalid when continuous gas pathways develop or when gas dissolution, exsolution, or volatilisation exerts a notable influence~~

[Added new content:] *The porous medium is assumed to remain nearly saturated. The liquid phase forms a continuous network and controls flow, while the gas phase is idealised as isolated, immobile bubbles that do not participate in advection. Hence, the degree of saturation $S_r$ is prescribed as a constant rather than solved as a state variable, leading to a single-phase liquid formulation in which residual gas influences the system only through the effective compressibility of the pore fluid–solid skeleton. This treatment is consistent with mass conservation under drained conditions, whereby the decrease in pore volume during consolidation is accommodated by outward Darcy drainage, so that the volumetric water content $\theta = nS_r$ evolves implicitly through the liquid-phase mass balance while $S_r$ is treated as approximately constant. This approximation is suitable for wet, low-permeability geomaterials where $S_r$ typically exceeds 0.8–0.9 and varies slowly relative to the consolidation timescale, such that deformation-driven transport dominates over transient unsaturated flow, as in compacted clay barriers and landfill liners under quasi-steady infiltration.*

*The formulation becomes invalid when drying fronts develop, when strong transient infiltration induces significant saturation changes, or when connected gas pathways form such that gas-phase transport is no longer negligible. Under these circumstances, prescribing $S_r$ is inappropriate and the single-phase assumption may fail (Zeng et al., 2011). Accordingly, the present modelling philosophy is to focus on deformation-driven transport under quasi-saturated conditions, rather than adopting thermodynamics-based unsaturated multi-field frameworks in which saturation, multiphase pore pressures and chemical potentials are fully coupled (Liu et al., 2025). This simplification is appropriate for a wide range of low-permeability geomaterials under near-saturated conditions, in which saturation evolves slowly relative to the consolidation timescale and gas connectivity remains limited, so that deformation-driven transport dominates over transient unsaturated flow processes.*

**[Line 131–145]**

**References**

Liu, J.H., Ma, T.S., Fu, J.H., Gao, J.J., Martyushev, D.A., Ranjith, P.G., 2025. Thermodynamics-based unsaturated hydro-mechanical-chemical coupling model for wellbore stability analysis in chemically active gas formations. Journal of Rock Mechanics and Geotechnical Engineering 17, 3644–3661. doi:https://doi.org/10.1016/j.jrmge.2024.09.024.

Zeng, Y.J., Su, Z.B., Wan, L., Wen, J., 2011. Numerical analysis of air-water-heat flow in unsaturated soil: Is it necessary to consider airflow in land surface models? Journal of Geophysical Research: Atmospheres 116. doi:https://doi.org/10.1029/2011JD015835.

---

## Author Comment (AC5)

**Response Statement to Community's Comments (CC5)**

Wang and Jeng

January 24, 2026

The authors thank the reviewer for the valuable comments. The manuscript has been revised by carefully considering all the comments. The changes are highlighted in the marked copy, and detailed responses to the reviewer's comments are provided below.

**Comment #CC5:**

*At minimum, include: (1) The PHREEQC database used, the key input blocks (KINETICS/EQUILIBRIUM PHASES/etc.), and where they are provided (supplement/Git repo). (2) OpenFOAM discretisation choices for transport (schemes, limiters), linear solvers/preconditioners, coupling tolerances, and convergence criteria (you mention a prescribed tolerance but not what it is). (3) Use a public repository plus archived case files. Without this, the ChemWindow contribution especially will be hard to credit.*

**Response:**

We thank the reviewer for highlighting the importance of reproducibility and transparency. The manuscript has been revised accordingly as follows.

1. **PHREEQC database and input blocks.** We now explicitly state that the thermodynamic database used is `phreeqc.dat` obtained from the official USGS PHREEQC distribution. The conventional PHREEQC input blocks (e.g. `SOLUTION`, `EQUILIBRIUM_PHASES`, and `KINETICS`) are encoded in OpenFOAM chemistry dictionaries specified by the entry `PhreeqcInputFile "phreeqcInput"`, following the `porousMedia4Foam` framework. A detailed description of the OpenFOAM–PHREEQC mapping and representative configuration options is now provided in Appendix C.

2. **OpenFOAM discretisation and solver settings.** A concise description of the discretisation schemes for solute transport, the linear solvers and preconditioners, and the coupling tolerances and convergence criteria has been added to Section 3.3. This includes the convective and diffusive schemes, the solver types and

preconditioners for the pressure, displacement and concentration equations, and the absolute and relative tolerances used to advance the HMC system.

3. **Public repository and archived case files.** A public repository containing the ChemWindow implementation, representative OpenFOAM chemistry dictionaries, and minimal working cases is currently being prepared and will be released upon acceptance of the manuscript. The repository will provide all configuration files and scripts required to reproduce the numerical results reported in this study.

[Added new content:] *Within the OpenFOAM framework adopted in this study, the governing equations were discretised using the finite volume method. Diffusive and dispersive fluxes were evaluated using the Gauss linear scheme, while the advective term of solute transport was discretised using a Gauss limitedLinear scheme with a limiter coefficient of 1 in order to suppress spurious oscillations. Temporal integration was performed using the implicit Euler method. The pressure, displacement and concentration equations were solved using the GAMG solver with a DILU preconditioner, with absolute and relative tolerances of $10^{-9}$ and zero, respectively. The coupled HMC system was advanced using a segregated outer-iteration strategy, and convergence was achieved when the residuals of all primary variables dropped below $10^{-6}$. Further implementation details are provided in the companion paper (Wang and Jeng, 2025).*

**[Line 76–80]**

*Geochemical reactions are executed through PHREEQC using the PhreeqcRM interface. In contrast to conventional PHREEQC workflows based on standalone input scripts, the present solver introduces a dictionary-based configuration layer within OpenFOAM. Keywords such as* `PhreeqcDataBase` *and* `PhreeqcInputFile` *are parsed at runtime and directly mapped to the corresponding PhreeqcRM function calls. Consequently, all reaction systems are fully configurable at runtime and no recompilation of the solver is required when modifying geochemical databases or reaction settings. This design establishes the solver as a configurable reactive-transport computing platform rather than a case-specific implementation.*

**Database binding and runtime mapping**

*The thermodynamic database is specified by the keyword* `PhreeqcDataBase`, *which is passed directly to the PhreeqcRM initialisation routine. Similarly, the entry* `PhreeqcInputFile` *defines an OpenFOAM dictionary that encodes the aqueous solution composition using standard PHREEQC* `SOLUTION` *syntax. At runtime, the PhreeqcRM interface converts the dictionary entries into the corresponding PHREEQC command strings and executes them internally via PhreeqcRM, ensuring full compatibility with the official PHREEQC syntax.*

**Reaction model configuration**

*Phase equilibrium and kinetic reactions are configured through the boolean flag* `activatePhaseEquilibrium`. *When enabled, mineral reactions are treated as equilibrium phases; otherwise, they are processed as kinetic reactions, reproducing the functionality of the PHREEQC blocks* `EQUILIBRIUM_PHASES` *and* `KINETICS`. *Time integration of kinetic reactions is controlled through the options* `cvODE` *and* `cvODETol`, *which activate the CVODE integrator and prescribe its relative tolerance, respectively. The keyword* `setComponentH2O` *determines whether water is treated as an explicit geochemical component.*

**Extensibility**

*All chemistry-related options are defined exclusively in OpenFOAM dictionaries and are translated to PhreeqcRM calls at runtime. Therefore, new reaction systems, databases, and kinetic models can be introduced by editing configuration files only, without any modification or recompilation of the solver source code.*

[Line 76–80]

[Line 258–259]

**References**

Wang, B.L., Jeng, D.S., 2025. Three-dimensional model for consolidation-induced solute transport in a nearly saturated porous medium. International Journal for Numerical and Analytical Methods in Geomechanics 49, 4436–4464. doi:https://doi.org/10.1002/nag.70070.

---

## Author Comment (AC6)

**Response Statement to Community's Comments (CC6)**

Wang and Jeng

January 24, 2026

The authors thank the reviewer for the valuable comments. The manuscript has been revised by carefully considering all the comments. The changes are highlighted in the marked copy, and detailed responses to the reviewer's comments are provided below.

**Comment #CC6:**

*I have some concerns about hybrid decisions. (1) You use element-based ADE transport, with a weighted molecular diffusion coefficient built from prescribed species fractions, while noting that fully coupled PhreeqcRM could update speciation and thereby Dm dynamically, but you don't do that in this study. (2) You also include a linear adsorption Kd term in the ADE, while stating that more detailed sorption could be handled by PHREEQC reaction terms. Please add a why this choice paragraph: is the goal robustness and speed? If so, say that explicitly and acknowledge the trade-off. Specify whether Kd is used in the case studies and, if yes, whether PHREEQC includes any overlapping sorption reactions.*

**Response:**

Thank you for raising this important point. We agree that the modelling choices regarding the treatment of diffusion coefficients and sorption processes must be clearly stated.

In this study, the species fractions $\omega_i$ used to compute the effective molecular diffusion coefficient $D_m = \sum_i \omega_i D_m^{(i)}$ are prescribed as constants. Although the numerical architecture can, in principle, be extended to dynamically update $\omega_i$ using cell-wise speciation results from a geochemical module, such functionality is *not activated in this work*. Dynamic coupling would require code-level modifications and recompilation whenever the reaction network is altered, which substantially reduces model reproducibility. Moreover, as $D_m$ is formulated as a weighted average of species-specific diffusion coefficients, its variation under typical speciation changes is expected to be limited and is unlikely to affect the principal chemo-mechanical trends examined here. We have revised the manuscript to explicitly state this rationale.

Regarding sorption, although the governing equation is presented in a general form that can accommodate a linear distribution coefficient $K_d$, *all simulations reported in this paper set $K_d = 0$; that is, no equilibrium sorption or retardation is considered.* Furthermore, no PHREEQC-based sorption reactions are activated in the present study, and therefore there is no overlap or double-counting between an ADE retardation term and geochemical reaction processes. The statement that PHREEQC can represent more complex sorption mechanisms has been clarified as a future extension and is not part of the current implementation.

[Deleted content:]

[Added new content:] *The PHREEQC reaction module is, in principle, capable of representing more complex sorption mechanisms (e.g. nonlinear Freundlich or Langmuir isotherms and competitive or kinetic adsorption–desorption), but these functionalities are reserved for future extensions and are not used in this work.*

**[Line 208–209]**

[Deleted content:] ~~Within the coupled hydro–mechanical–chemical framework, the relative fractions of aqueous species $\omega_i$ associated with each element can be dynamically obtained or prescribed as constants. When executed in a fully coupled mode with PhreeqcRM, the speciation module automatically determines species concentrations in each cell and time step, allowing $\omega_i = Y_i/Y_{\text{element}}$ to evolve with local chemistry. This allows the effective diffusion coefficient $D_m = \sum_i \omega_i D_m^{(i)}$ to vary consistently with reaction, pH, and ionic strength. The species-specific diffusion coefficients $D_m^{(i)}$ are retrieved directly from the Phreeqc database via the `-diffusion_coefficient` entries, ensuring a thermodynamically consistent parameterisation for both static and dynamic simulations.~~

[Added new content:] *Within the coupled hydro–mechanical–chemical framework adopted in this study, the relative fractions of aqueous species $\omega_i$ associated with each element are prescribed as constant parameters, and the effective molecular diffusion coefficient is evaluated as $D_m = \sum_i \omega_i D_m^{(i)}$, where the species-specific diffusion coefficients $D_m^{(i)}$ are obtained from the PHREEQC database through the `-diffusion_coefficient` entries. Although the present numerical architecture can, in principle, be extended to dynamically update $\omega_i$ using cell-wise speciation results from a geochemical module, such functionality is* not activated in this work*, because dynamic coupling would require code-level modifications and recompilation whenever the reaction network is altered, substantially reducing model reproducibility; moreover, as $D_m$ is formulated as a weighted average of species diffusion coefficients, its variation under typical speciation changes is expected to be limited and is unlikely to affect the principal chemo-mechanical trends investigated here.*

**[Line 208–209]**

---

## Author Comment (AC7)

**Response Statement to Community's Comments (CC7)**

**Wang and Jeng**

**January 24, 2026**

The authors thank the reviewer for the valuable comments. The manuscript has been revised by carefully considering all the comments. The changes are highlighted in the marked copy, and detailed responses to the reviewer's comments are provided below.

**Comment #CC7:**

*You model landfill-liner-like consolidation with chemistry. There are some questions with (1) the mechanical model (elastic vs elasto-plastic/creep typical in clays), (2) the lack of reaction-induced porosity/permeability evolution (I know you already acknowledge this which is good, but it reduces interpretability of long-term plume changes), (3) boundary conditions (drainage, injection as fluxes at multiple points) and whether they represent a realistic exposure. Consider adding a short justification for the constitutive law used and explicitly call the case studies "demonstration problems" rather than predictive landfill designs. Add one sentence in the Conclusions about what would change when property feedbacks are included (directional expectation is fine, but label it as expectation).*

*Also some minor comments below:*

*(1) In benchmark 2 you say early-time deviations come from transport formulation and boundary representations; please add one sentence indicating which boundary treatment differs (Dirichlet/Neumann/mixed; inlet formulation), and whether refinement improves it.*

*(2) Consistently distinguish "hydraulic conductivity K" vs "permeability k" (the symbol list emphasises K).*

*(3) Injection strengths are given as fluxes at J1–J3; ensure it's always explicit whether that is imposed as boundary flux ($mol/(m^2 \cdot s)$) and over what area.*

*(4) Computational performance section: helpful, but reviewers will ask for scaling beyond 4 MPI ranks and/or mesh size dependence. Even a small strong-scaling plot (1,2,4,8,16) on one case would help.*

*(5) Consider moving the limitations to earlier in the discussion.*

**Response:**

We thank the reviewer for the constructive and insightful comments, which have helped us to better clarify the scope, assumptions, and applicability of the proposed framework. The manuscript has been revised accordingly. Our detailed responses are provided below.

**Major comments**

1. A justification of the small-strain elastic constitutive assumption has been added to the Conclusions. We now explicitly state that the formulation is intended for scenarios in which deformations remain limited and significant yielding or creep does not occur, and that elasto-plastic or visco-plastic constitutive laws would be required for long-term simulations involving large settlements or time-dependent deformation of clay liners.

2. We have added a statement in the Conclusions clarifying that reaction-induced porosity–permeability feedbacks are not considered in the present study. We further note that incorporating such feedbacks is *expected* to intensify preferential flow and increase the spatial heterogeneity of solute plumes over long time scales. This statement is explicitly framed as a qualitative expectation rather than a quantified prediction.

3. We have clarified that the boundary conditions and source configurations are intended as demonstration problems designed to explore coupled hydro–mechanical–chemical mechanisms, rather than predictive landfill-scale exposure scenarios. In particular, we now explicitly state that the injection patterns represent localised defects or preferential leakage pathways, whereas spatially uniform large-scale leakage is not considered representative of engineered barrier systems. Owing to the absence of site-specific field data, the simulations are therefore framed as scenario-based analyses rather than field-calibrated predictions. This clarification has been added at the end of the boundary-condition subsection.

[Added new content:] *This study presents a multi-dimensional HMC modelling framework by embedding a flexible geochemical module into an existing hydro-mechanical solver, enabling the consistent treatment of multi-component aqueous species and mineral reactions in near-saturated deformable porous media. The formulation is developed for small-strain, predominantly elastic behaviour, applicable to scenarios in which deformations remain limited and the soil skeleton does not undergo significant yielding or creep. For long-term conditions involving large settlements or plastic deformation of soft clays, an elasto-plastic or visco-plastic constitutive description, together with reaction-induced porosity–permeability feedbacks, would be required. Incorporating these effects is expected to intensify preferential flow and increase the spatial heterogeneity of solute plumes over time. These developments, together with quantitative benchmarking against laboratory and field data, will be pursued to further strengthen the predictive capability of the framework.*

**[Line 258–259]**

[Added new content:] *It is emphasised that the imposed concentration boundary conditions are adopted as controlled demonstration settings to explore the sensitivity of coupled hydro-mechanical–chemical responses to source strength and drainage configuration. The leakage cases considered correspond to localised defects or preferential pathways rather than spatially uniform barrier failure, which is not regarded as realistic for engineered containment systems. In the absence of site-specific investigation data, the present simulations should therefore be interpreted as scenario-based analyses aimed at mechanistic understanding, rather than predictive landfill exposure assessments.*

**[Line 258–259]**

Minor comments:

1. We thank the reviewer for this comment. In Benchmark #2, both the reference solver and the present framework prescribe inlet concentration using fixedValue (Dirichlet-type) boundary conditions. However, the effective advective boundary fluxes differ because the reference implementation evaluates advection using the face-based volumetric flux $\phi$, whereas the present model employs a cell-centred Darcy-based velocity field that is interpolated to the boundaries. This structural difference leads to irreducible early-time discrepancies that cannot be eliminated by mesh or time-step refinement.

2. This has now been corrected throughout the manuscript.

3. We have clarified that $J_1$–$J_3$ are imposed as prescribed boundary mass fluxes with units of $\mathrm{mol\,m^{-2}\,s^{-1}}$ over the corresponding inlet areas along the top boundary. The description of the top boundary conditions has been revised accordingly.

4. We agree that this would strengthen the manuscript. A strong-scaling test (1, 2, 4, 8 and 16 MPI ranks) has now been added for a representative case, and the results are reported in the revised computational performance section.

5. The limitations associated with the elastic constitutive law and the absence of reaction-induced property feedbacks are now discussed in the opening paragraph of the Conclusions to ensure that the scope and applicability of the model are clearly stated before the main findings are summarised.

[Added new content:] *Along the top boundary, three contaminant inlets (depicted by blue arrows) are prescribed as boundary molar fluxes $J_1$–$J_3$ with units of $\mathrm{mol/(m^2 \cdot s)}$. Each inlet is applied uniformly over a top-boundary patch of width $4$ m (i.e., an inlet area $A_i = 4$ m $\times$ $0.02$ m, where $0.02$ m is the out-of-plane thickness of the quasi-2D OpenFOAM model). The intervening top-boundary segments are assigned zero solute flux. The entire top surface is subjected to a uniform mechanical load.*

**[Line 258–259]**

[Added new content:] *To evaluate the parallel efficiency of the developed solver, the wall time required for a five-year simulation is analysed as a function of the number of MPI processes. Figure 1 presents the measured wall time together with the corresponding speedup, which is defined relative to the baseline case with two MPI processes as Speedup = $T_2/T_N$.*

*As the number of MPI processes increases from 2 to 16, the wall time decreases substantially from approximately $3.5 \times 10^4$ s to below $5 \times 10^3$ s, demonstrating a marked improvement in computational efficiency. The speedup increases monotonically over the same range and remains close to linear scaling, indicating that the developed solver is well parallelised and can effectively exploit distributed-memory computing for the problem size considered. Overall, these results confirm satisfactory strong-scaling performance up to 16 MPI processes for the five-year simulation.*

[Figure]

Figure 1: Wall time and speedup versus number of MPI processes.

**[Line 258–259]**

---

## Author Comment (AC8)

**Response Statement to Referee's Comments (RC1)**

**Wang and Jeng**

**January 24, 2026**

The authors thank the reviewer for the valuable comments. The manuscript has been revised by carefully considering all the comments. The changes are highlighted in the marked copy, and detailed responses to the reviewer's comments are provided below.

**Comment #RC1:**

*E.g. validation : Line 280 - information regarding mesh, parameters, boundary conditions and similar are not reported. This circmstance makes the reproducibility of the results a major concern.*

**Response:**

We thank the reviewer for highlighting this important issue regarding reproducibility. We acknowledge that essential numerical information was insufficiently reported in the previous version, which could hinder the reproducibility of the results.

To address this concern, a new subsection entitled *"Numerical stability"* has been added to the manuscript. This subsection provides detailed information on the adopted mesh characteristics, time-step sizes for both the hydro-mechanical and chemical processes, solver configuration, convergence tolerances, and the sensitivity of the numerical solution to these numerical settings.

In addition, the boundary and initial conditions, together with the key material parameters used in the validation cases, are now explicitly reported in Section 4.2. With these additions, the numerical setup required to reproduce the presented results is now documented in sufficient detail in the revised manuscript.

---

## Author Comment (AC9)

**Response Statement to Referee's Comments (RC2)**

Wang and Jeng

January 24, 2026

The authors thank the reviewer for the valuable comments. The manuscript has been revised by carefully considering all the comments. The changes are highlighted in the marked copy, and detailed responses to the reviewer's comments are provided below.

**Comment #RC2:**

*please better clarify how saturation can remain constant upon loading and compaction.*
*why wasn't the water content considered instead?*

**Response:**

We thank the reviewer for this insightful comment. We agree that, in deforming porous media, the physically conserved quantity is the volumetric water content $\theta = nS_r$, and that prescribing a constant degree of saturation without clarification may appear inconsistent with mass conservation.

In the present work, drained loading conditions are assumed, such that the reduction in pore volume during consolidation is primarily accommodated by outward Darcy drainage through the permeable boundaries. Under these conditions, the volumetric water content is not prescribed as constant but is allowed to evolve implicitly through the liquid-phase mass balance and boundary fluxes. For near-saturated geomaterials, where gas connectivity remains limited and no significant desaturation occurs, the degree of saturation may therefore be treated as approximately constant without violating mass conservation.

This modelling assumption and its physical basis have now been explicitly clarified in the manuscript. We further note that this formulation becomes invalid when drainage is impeded or when significant unsaturated flow develops, in which case the degree of saturation (or volumetric water content) must be treated as an independent state variable within a fully coupled multiphase framework.

[Added new content:] *The porous medium is assumed to remain nearly saturated. The liquid phase forms a continuous network and controls flow, while the gas phase is idealised as isolated, immobile bubbles that do not*

*participate in advection. Hence, the degree of saturation $S_r$ is prescribed as a constant rather than solved as a state variable, leading to a single-phase liquid formulation in which residual gas influences the system only through the effective compressibility of the pore fluid–solid skeleton. This treatment is consistent with mass conservation under drained conditions, whereby the decrease in pore volume during consolidation is accommodated by outward Darcy drainage, so that the volumetric water content $\theta = nS_r$ evolves implicitly through the liquid-phase mass balance while $S_r$ is treated as approximately constant. This approximation is suitable for wet, low-permeability geomaterials where $S_r$ typically exceeds 0.8–0.9 and varies slowly relative to the consolidation timescale, such that deformation-driven transport dominates over transient unsaturated flow, as in compacted clay barriers and landfill liners under quasi-steady infiltration.*

**[Line 258–259]**

---

## Author Comment (AC10)

**Response Statement to Referee's Comments (RC3)**

Wang and Jeng

January 24, 2026

The authors thank the reviewer for the valuable comments. The manuscript has been revised by carefully considering all the comments. The changes are highlighted in the marked copy, and detailed responses to the reviewer's comments are provided below.

**General Comment:**

*The work addresses a well-recognized gap in reactive transport modelling: the integration of mechanical deformation with multicomponent geochemistry. The manuscript is generally well written and logically organized, and it fits well within the scope of coupled processes in hydrology and earth sciences.*

**Response (General Comment):**

We thank the reviewer for the positive and encouraging assessment of our work. We appreciate the recognition that the manuscript addresses a well-recognised gap in reactive transport modelling by integrating mechanical deformation with multicomponent geochemistry, and that it aligns well with the scope of coupled-process studies in hydrology and earth sciences.

We have carefully reviewed the manuscript and implemented targeted revisions to further clarify the coupling framework and to improve the overall clarity and organisation of the presentation, while maintaining a clear focus on the physical and chemical mechanisms underlying the proposed model.

**Comment #RC3-1:**

*It would be helpful to more explicitly articulate what new capabilities are enabled here beyond coupling Open-FOAM with PhreeqcRM. For example, a short comparison paragraph contrasting this framework with TOUGHRE-ACT or COMSOL approaches would clarify this.*

**Response:**

We thank the reviewer for this helpful suggestion. We have added a short comparison paragraph clarifying that, beyond a technical coupling of OpenFOAM and PhreeqcRM, the proposed framework enables equation-level formulation and control of fully coupled hydro–mechanical–chemical processes, in which mechanical deformation is explicitly represented as one of the driving mechanisms influencing reactive transport.

[Added new content:] *While established platforms such as TOUGHREACT and multiphysics implementations in COMSOL provide comprehensive tools for reactive transport modelling, they are not specifically developed to investigate deformation-driven reactive transport in near-saturated, deformable porous media within a unified continuum-mechanics framework. The strength of the present framework lies in its research-oriented numerical architecture, which allows the governing equations, coupling structure, and solution strategy of hydro–mechanical–chemical processes to be explicitly formulated and systematically controlled at the equation level. This capability is central to investigating deformation-driven reactive transport in near-saturated deformable porous media, which is the focus of this study.*

[Line 258–259]

**Comment #RC3-2:**

*The assumption of constant saturation (which may not be true in vadose zones) is central to several conclusions. I wonder what the physical interpretation of "near-saturated" is and how sensitive the results are to small variations in saturation.*

**Response:**

Thank you for this insightful comment. We agree that the assumption of constant saturation requires careful physical interpretation, particularly in vadose-zone settings.

In the revised manuscript, we clarify that "near-saturated" conditions refer to a regime in which the liquid phase forms a continuous network that governs flow, while the gas phase is present only as isolated, immobile bubbles that do not participate in advective transport. Under these conditions, saturation may be treated as approximately constant, and the influence of residual gas is primarily reflected through its contribution to the effective compressibility of the pore fluid–solid skeleton. This justifies the adoption of a single-phase liquid formulation within the specified range of applicability.

We have also explicitly stated the limitations of this assumption, noting that it becomes inappropriate when significant drying, strong transient infiltration, or connected gas pathways develop, in which case a fully unsaturated multiphase formulation would be required.

With respect to sensitivity, additional analyses within the tested saturation range ($S_r = 0.8$–$1.0$) indicate that variations in saturation significantly affect the magnitude and rate of hydro-mechanical responses, such as excess pore pressure dissipation and stabilisation time, while the ultimate mechanical deformation (final settlement) remains relatively insensitive. In contrast, the chemical response exhibits higher sensitivity: decreasing saturation leads to increased aqueous solute concentrations and an expanded plume extent, associated with changes in advective transport intensity and effective aqueous volume. Importantly, although saturation influences response magnitudes, the governing mechanisms and comparative trends remain consistent across the investigated saturation range.

[Added new content:] *The porous medium is assumed to remain nearly saturated. The liquid phase forms a continuous network and controls flow, while the gas phase is idealised as isolated, immobile bubbles that do not participate in advection. Hence, the degree of saturation $S_r$ is prescribed as a constant rather than solved as a state variable, leading to a single-phase liquid formulation in which residual gas influences the system only through the effective compressibility of the pore fluid–solid skeleton. This treatment is consistent with mass conservation under drained conditions, whereby the decrease in pore volume during consolidation is accommodated by outward Darcy drainage, so that the volumetric water content $\theta = nS_r$ evolves implicitly through the liquid-phase mass balance while $S_r$ is treated as approximately constant. This approximation is suitable for wet, low-permeability geomaterials where $S_r$ typically exceeds $0.8$–$0.9$ and varies slowly relative to the consolidation timescale, such that deformation-driven transport dominates over transient unsaturated flow, as in compacted clay barriers and landfill liners under quasi-steady infiltration.*

*The formulation becomes invalid when drying fronts develop, when strong transient infiltration induces significant saturation changes, or when connected gas pathways form such that gas-phase transport is no longer negligible. Under these circumstances, prescribing $S_r$ is inappropriate and the single-phase assumption may fail (Zeng et al., 2011). Accordingly, the present modelling philosophy is to focus on deformation-driven transport under quasi-saturated conditions, rather than adopting thermodynamics-based unsaturated multi-field frameworks in which saturation, multiphase pore pressures and chemical potentials are fully coupled (Liu et al., 2025). This simplification is appropriate for a wide range of low-permeability geomaterials under near-saturated conditions, in which saturation evolves slowly relative to the consolidation timescale and gas connectivity remains limited, so that deformation-driven transport dominates over transient unsaturated flow processes.*

**[Line 258–259]**

[Deleted content:]

[Added new content:] *Overall, the results indicate that saturation variations exert a stronger control on chemical concentrations and plume geometry than on the final mechanical deformation. Although saturation influences*

*response magnitudes, the governing mechanisms and comparative trends remain consistent across $S_r$ 0.8-1.0.*

**[Line 258–259]**

[Deleted content:]

[Added new content:] *Saturation plays a key role in both hydro-mechanical responses and reactive-solute transport, with a more pronounced influence on chemical concentrations and plume evolution.*

**[Line 258–259]**

**Comment #RC3-3:**

*The manuscript excludes reaction-induced changes in porosity and permeability. The authors should more clearly discuss how this assumption may bias long-term predictions.*

**Response:**

Thank you for this insightful comment. In the revised manuscript, we have expanded the discussion to more clearly explain how neglecting reaction-induced porosity and permeability evolution may influence long-term predictions.

Specifically, we now clarify that excluding these feedback mechanisms may lead to a systematic bias towards smoother flow fields and less heterogeneous concentration distributions in reactive environments. Reaction-driven changes in pore structure, such as mineral dissolution or precipitation, are known to locally modify permeability and porosity, potentially enhancing preferential flow pathways and amplifying plume irregularity over long time scales. By neglecting these effects, the present framework may therefore underestimate the degree of spatial variability and channelling in long-term solute transport.

Accordingly, the model results should be interpreted as providing conservative estimates of plume spreading and migration patterns under reactive conditions. This limitation, together with its implications for long-term predictions, is now explicitly discussed in the revised manuscript.

[Added new content:] *For long-term conditions involving large settlements, plastic deformation of soft clays, or pronounced geochemical alteration, a more comprehensive framework would be required, incorporating elasto-plastic or visco-plastic constitutive behaviour together with reaction-driven porosity–permeability feedbacks. These processes are expected to intensify preferential flow pathways and enhance solute plume heterogeneity, such that neglecting them may lead to smoother transport patterns and an underestimation of plume spreading in reactive environments. Future work will focus on integrating these feedbacks and on quantitative benchmarking against laboratory and field observations to further strengthen the predictive capability of the framework.*

**Comment #RC3-4:**

*The treatment of molecular diffusion uses a weighted-average diffusion coefficient. I wonder how these weights are selected in practice.*

**Response:**

The weighting factors $\omega_i$ are not arbitrarily selected. They are obtained from the initial geochemical speciation calculations and represent the relative fractions of aqueous species at the start of the simulations. In the present implementation, these fractions are prescribed as constant parameters for clarity and reproducibility. The effective molecular diffusion coefficient is then evaluated as a weighted average of species-specific diffusion coefficients retrieved from the PHREEQC database. A clarification has been added to the manuscript to make this procedure explicit.

[Added new content:] *Within the coupled hydro–mechanical–chemical framework adopted in this study, the relative fractions of aqueous species $\omega_i$, obtained from the initial geochemical speciation calculations, are prescribed as constant parameters.*

[Line 258–259]

**References**

Liu, J.H., Ma, T.S., Fu, J.H., Gao, J.J., Martyushev, D.A., Ranjith, P.G., 2025. Thermodynamics-based unsaturated hydro-mechanical-chemical coupling model for wellbore stability analysis in chemically active gas formations. Journal of Rock Mechanics and Geotechnical Engineering 17, 3644–3661. doi:https://doi.org/10.1016/j.jrmge.2024.09.024.

Zeng, Y.J., Su, Z.B., Wan, L., Wen, J., 2011. Numerical analysis of air-water-heat flow in unsaturated soil: Is it necessary to consider airflow in land surface models? Journal of Geophysical Research: Atmospheres 116. doi:https://doi.org/10.1029/2011JD015835.